# Strategies found not to be suitable for stabilizing high steroid hydroxylation activities of CYP450 BM3-based whole-cell biocatalysts

Carolin Bertelmann[1,2], Bruno Bühler[1,2]*

1 Department of Solar Materials Biotechnology, Helmholtz Centre for Environmental Research GmbH - UFZ, Leipzig, Saxony, Germany, 2 Department of Microbial Biotechnology, Helmholtz Centre for Environmental Research GmbH - UFZ, Leipzig, Saxony, Germany

* bruno.buehler@ufz.de

**Data Availability Statement:** All relevant data are within the manuscript and its Supporting information files.

## Abstract

The implementation of biocatalytic steroid hydroxylation processes plays a crucial role in the pharmaceutical industry due to a plethora of medicative effects of hydroxylated steroid derivatives and their crucial role in drug approval processes. Cytochrome P450 monooxygenases (CYP450s) typically constitute the key enzymes catalyzing these reactions, but commonly entail drawbacks such as poor catalytic rates and the dependency on additional redox proteins for electron transfer from NAD(P)H to the active site. Recently, these bottlenecks were overcome by equipping *Escherichia coli* cells with highly active variants of the self-sufficient single-component CYP450 BM3 together with hydrophobic outer membrane proteins facilitating cellular steroid uptake. The combination of the BM3 variant KSA14m and the outer membrane pore AlkL enabled exceptionally high testosterone hydroxylation rates of up to 45 U $g_{CDW}^{-1}$ for resting (i.e., living but non-growing) cells. However, a rapid loss of specific activity heavily compromised final product titers and overall space-time yields. In this study, several stabilization strategies were evaluated on enzyme-, cell-, and reaction level. However, neither changes in biocatalyst configuration nor variation of cultivation media, expression systems, or inducer concentrations led to considerable improvement. This qualified the so-far used genetic construct pETM11-ksa14m-alkL, M9 medium, and the resting-cell state as the best options enabling comparatively efficient activity along with fast growth prior to biotransformation. In summary, we report several approaches not enabling a stabilization of the high testosterone hydroxylation rates, providing vital guidance for researchers tackling similar CYP450 stability issues. A comparison with more stable natively steroid-hydroxylating CYP106A2 and CYP154C5 in equivalent setups further highlighted the high potential of the investigated CYP450 BM3-based whole-cell biocatalysts. The immense and continuously developing repertoire of enzyme engineering strategies provides promising options to stabilize the highly active biocatalysts.

**Funding:** The study was financially supported by Bayer AG (no grant number). Website: https://www.bayer.com BB aquired the funding and CB was paid from the funding. Bayer AG released the submitted paper for publication. The funders participated in project design, but had no role in study design, data collection and analysis, or preparation of the manuscript.

**Competing interests:** The authors have declared that no competing interests exist.

## Introduction

Steroid hydroxylations hold significant relevance for the pharmaceutical industry due to the vast scope of hydroxylated steroid derivatives displaying specific pharmacological activities, e.g., anti-inflammatory, contraceptive, or immunosuppressive effects, but, being formed in the liver, also adverse effects making them relevant for drug approval processes [1, 2]. Desired reactions typically are accomplished using cytochrome P450 monooxygenases (CYP450s) which offer an enormous biocatalytic versatility regarding substrate, regio-, and stereoselectivity [1, 3]. Despite their impressive synthetic potential, native steroid-hydroxylating CYP450s suffer from drawbacks that often hamper their biotechnological use. Commonly observed limitations include poor catalytic activity, instability in isolated form, as well as insufficient selectivity [4–6]. They require an external electron donor, more specifically NAD(P)H, which supplies the electrons necessary for reductive $O_2$ activation [7, 8]. The majority of CYP450s furthermore depends on specific redox proteins enabling the electron transfer to the heme iron in the active site [4, 5, 7, 9]. Respective subunits need to be expressed in a stable and active form and in the optimal ratio [10, 11], with a crucial influence on biotransformation efficiency [12, 13]. Biocatalytic steroid hydroxylation therefore is expected to benefit from self-sufficient systems such as CYP450 BM3 from *Bacillus megaterium* ATCC14581, in which both the heme-containing monooxygenase and FAD/FMN-containing reductase functionalities are combined in a single polypeptide [14, 15].

In our previous research, we applied highly active BM3 variants [16] in living *Escherichia coli* cells for testosterone hydroxylation [17]. Typically, whole-cell biocatalysts represent the preferred format for CYP450 catalysis, since they allow for enzyme resynthesis and stabilization, cofactor recycling, and reactive oxygen species (ROS) degradation [12, 18, 19]. In order to relieve limiting cellular uptake of the substrate testosterone, the hydrophobic outer membrane pore AlkL from *Pseudomonas putida* GPo1 was introduced. The genes encoding different BM3 variants and AlkL were co-expressed from a bicistronic operon under control of the *lacI*-P$_{T7}$-based regulatory system (S1A Fig) on the pETM11 vector, giving the recombinant strain *E. coli* BL21-Gold(DE3)_pETM11-ksa14m-alkL. Resulting whole-cell biocatalysts showed exceptionally high specific testosterone hydroxylation rates [17], which exceeded those reported for other testosterone-hydroxylating CYP450 systems by a factor of up to 264 [20, 21]. Resting (i.e., metabolically active but non-growing) *E. coli* cells harboring AlkL and the BM3 variant KSA14m displayed the highest specific activity (45 U $g_{CDW}^{-1}$), paving the way towards unprecedented product titers and space-time yields for steroid hydroxylation processes. However, poor operational stability of this whole-cell biocatalyst has been confirmed [22]. A systematic evaluation of kinetics and stability identified limited substrate availability due to poor testosterone solubility, product inhibition, and enzyme instability as critical factors. The latter appeared to involve enzyme inactivation rather than degradation and occurred independently from the presence of substrate and products, revealing an inherent instability of KSA14m possibly *via* uncoupling [22]. Engineered BM3 variants often suffer from poor coupling between product formation and NAD(P)H consumption, which typically involves ROS formation [16, 23]. Such oxidative stress can harm both protein function and cell physiology [24–26] and thus is detrimental for biocatalytic reactions relying on active metabolism [27]. All in all, especially KSA14m instability constrained both process efficiency and durability and consequently constitutes an optimization target for improving testosterone hydroxylation using the reported BM3 variant KSA14m.

Numerous strategies on enzyme, cell, and reaction level have proven successful for the stabilization of oxygenase-based catalysis [27] and thus constitute targets of this study for the optimization of KSA14m-based testosterone hydroxylation in *E. coli*. Whole-cell biocatalyst

performance relies on the cellular machinery providing energy and reduction equivalents for the synthesis of functional enzymes and the reaction itself. In this work's context, reactions catalyzed by the BM3 variant KSA14m are linked to cellular metabolism by the redox cofactor NADPH. Since the metabolic activity necessary for energy and redox equivalent supply is closely connected to the physiological state of cells, the cultivation medium and conditions are decisive for biocatalyst performance [28]. Resting *E. coli* cells, as employed in our previous studies [17, 22], have displayed superior whole-cell activities for redox biocatalysis compared to growing cells [29, 30]. However, as this condition is effectuated by the lack or depletion of, e.g., nitrogen sources in the medium [31], this biocatalyst format often is associated with poor stability [30, 32]. Interestingly, specific activities of magnesium-limited resting cells have been observed as more stable than those of nitrogen-deprived cells, highlighting the type of nutrient starvation as a possible stabilizing aspect for evaluation [33]. The application of growing instead of resting cells also can improve biocatalyst stability due to efficient (re-)synthesis of recombinant enzymes such as oxygenases [18] as well as more efficient handling of biocatalysis-related burdens like oxidative stress and product inhibition [30, 34]. Furthermore, adjusting oxygenase expression to a level that can be managed by cellular metabolism has proven beneficial for whole-cell biocatalyst stabilization [27]. To meet this purpose, fine-tunings of gene expression [35, 36] and cultivation media [28] represent promising strategies. Also the use of pseudomonads as microbial hosts may promote functional CYP450 synthesis, as, in contrast to *E. coli*, they feature intrinsic CYP450 genes and thus more efficient heme synthesis. Lastly, engineering the substrate binding pocket of CYP450s has been reported to considerably increase coupling efficiencies [37–39] and consequently may reduce oxidative stress and improve enzyme stability.

In this study, we set out to stabilize biocatalytic testosterone hydroxylation with *E. coli* cells harboring the BM3 variant KSA14m and AlkL as biocatalyst. Their performance was evaluated in different physiological states, cultivation media, and with various gene expression systems as well as inducer concentrations. *Pseudomonas taiwanensis* VLB120_Strep was investigated as alternative host. Further, KSA14m was engineered based on amino acid substitutions previously found to improve the coupling efficiency of a propane-hydroxylating BM3 variant [37]. Lastly, we compared specific activities and operational stability of *E. coli* BL21-Gold(DE3)_pETM11-ksa14m-alkL with those of cells carrying the natively steroid-hydroxylating oxygenases CYP106A2 and CYP154C5.

## Materials and methods

### Chemicals, gene synthesis, and oligonucleotides

2β- and 15β-hydroxytestosterone were obtained from Steraloids Inc. (Newport, RI, USA). All other chemicals were purchased from AppliChem (Darmstadt, Germany), Carl Roth (Karlsruhe, Germany), Chemsolute (Renningen, Germany), or Sigma-Aldrich (Steinheim, Germany) in the highest purity available. Custom synthesized genes and oligonucleotides were acquired from Eurofins (Ebersberg, Germany).

### Generation of recombinant bacterial strains

Microbial strains and plasmids used in this work are listed in Table 1. *E. coli* DH5α was used for cloning purposes, whereas *E. coli* BL21-Gold(DE3) or *P. taiwanensis* VLB120_Strep were used for expression and biotransformation studies. Electro-competent *E. coli* and *P. taiwanensis* cells were prepared as reported earlier [40, 41]. Plasmids were introduced *via* electroporation (2500 V, Eppendorf Eporator®, Hamburg, Germany).

**Table 1. Bacterial strains and plasmids used in this study.**

| Strain or plasmid | Characteristics | Host organisms in this study | Reference |
|---|---|---|---|
| **Strains** | | | |
| *E. coli* DH5α | *sup*E44 Δ*lac*U169(Φ80 *lacZ*ΔM15) *hsd*R17 *rec*A1 *end*A1 *gyr*A96 *thi*-1 *rel*A1 | | [42] |
| *E. coli* BL21-Gold(DE3) | F⁻ *ompT hsdS*(r_B⁻ m_B⁻) *dcm*⁺ *Tet*ᴿ *gal*λ(DE3) *endA* Hte | | Agilent Technologies Santa Clara, USA |
| *P. taiwanensis* VLB120_Strep | solvent-tolerant, styrene-degrading bacterium, isolated from forest soil, Strepᴿ on megaplasmid pSty (encodes solvent-tolerance genes) | | [43] |
| **Plasmids** | | | |
| pETM11 | pMB1 ori, *lac*-regulatory system (*lacI*, $P_{T7}$), Kmᴿ, pETM11 RBS | | EMBL Vector collection, Heidelberg, Germany |
| pCOM10 | broad-host-range expression vector (pRO1600 and ColE1 ori), *alk*-regulatory system (*alkS*, $P_{alkB}$), Kmᴿ, pCOM10 RBS | | [44] |
| pCOM10_lac | pCOM10 derivative, *lac*-regulatory system (*lacI*, $P_{lacUV5}$) | | [45] |
| pCOM10_tac | pCOM10 derivative, *lac*-regulatory system (*lacI*ᑫ, $P_{tac}$) | | [46] |
| pACYC_Duet1 | P15A ori, *lac*-regulatory system (*lacI*, 2x $P_{T7}$), Cmᴿ, pACYC RBS | | Merck KGaA, Darmstadt, Germany |
| pETM11-ksa14m-alkL | pETM11 derivative with 6xHis-tagged BM3 mutant gene *ksa14m* and *alkL* gene from *P. putida* GPo1 | *E. coli* BL21-Gold(DE3) | [17] |
| palk-ksa14m-alkL | pCOM10 derivative with 6xHis-tagged BM3 mutant gene *ksa14m* and *alkL* gene from *P. putida* GPo1 | *E. coli* BL21-Gold(DE3); *P. taiwanensis* VLB120_Strep | This study |
| plac-ksa14m-alkL | pCOM10_lac derivative with 6xHis-tagged BM3 mutant gene *ksa14m* and *alkL* gene from *P. putida* GPo1 | *E. coli* BL21-Gold(DE3); *P. taiwanensis* VLB120_Strep | This study |
| ptac-ksa14m-alkL | pCOM10_tac derivative with 6xHis-tagged BM3 mutant gene *ksa14m* and *alkL* gene from *P. putida* GPo1 | *P. taiwanensis* VLB120_Strep | This study |
| plac-ksa14m | pCOM10_lac derivative with 6xHis-tagged BM3 mutant gene *ksa14m* | *P. taiwanensis* VLB120_Strep | This study |
| plac-ksa14m-fhuAΔ1–160 | pCOM10_lac derivative with 6xHis-tagged BM3 mutant gene *ksa14m* and *fhuAΔ1–160* gene from *E. coli* MG1655 | *P. taiwanensis* VLB120_Strep | This study |
| plac-ksa14m-todX | pCOM10_lac derivative with 6xHis-tagged BM3 mutant gene *ksa14m and todX* gene from *P. putida* F1 | *P. taiwanensis* VLB120_Strep | This study |
| pETM11-ksa14m(M82A)-alkL | pETM11 derivative with 6xHis-tagged BM3 mutant gene *ksa14m* carrying M82A mutation and *alkL* gene from *P. putida* GPo1 | *E. coli* BL21-Gold(DE3) | This study |
| pETM11-ksa14m(L188P)-alkL | pETM11 derivative with 6xHis-tagged BM3 mutant gene *ksa14m* carrying L188P mutation and *alkL* gene from *P. putida* GPo1 | *E. coli* BL21-Gold(DE3) | This study |
| pETM11-ksa14mutations-alkL | pETM11 derivative with 6xHis-tagged BM3 mutant gene *ksa14m* carrying further mutations (L52I, A184V, I366V, G443A, D698G) and *alkL* gene from *P. putida* GPo1 | *E. coli* BL21-Gold(DE3) | This study |
| pET154-camAB | pETM11 derivative with codon-optimized 6xHis-tagged *cyp154c5* gene from *Nocardia farcinica* IFM 10152, *camA* and *camB* genes from *P. putida* DSM50198 | *E. coli* BL21-Gold(DE3) | This study |
| pET154-camAB-alkL | pET154c5-camAB derivative with *alkL* gene | *E. coli* BL21-Gold(DE3) | This study |
| pACYC-camAB | pACYC derivative with *camA* and *camB* genes from *P. putida* DSM50198 | *E. coli* BL21-Gold(DE3) | This study |
| ptac154 | pCOM10_tac derivative with codon-optimized *cyp154c5* gene from *Nocardia farcinica* IFM 10152 | *E. coli* BL21-Gold(DE3) | This study |
| ptac154A | ptac154 derivative with *alkL* gene from *P. putida* GPo1 | *E. coli* BL21-Gold(DE3) | This study |
| ptac106 | pCOM10_tac derivative with codon-optimized *cyp106a2* from *B. megaterium* ATCC 13368 | *E. coli* BL21-Gold(DE3) | This study |
| ptac106A | ptac106 derivative with *alkL* gene from *P. putida* GPo1 | *E. coli* BL21-Gold(DE3) | This study |
| pET154 | pETM11 derivative with codon-optimized *cyp154c5* gene from *N. farcinica* IFM 10152 | *E. coli* BL21-Gold(DE3) | This study |
| pET154A | pET154 derivative with *alkL* gene from *P. putida* GPo1 | *E. coli* BL21-Gold(DE3) | This study |
| pET106 | pETM11 derivative with codon-optimized *cyp106a2* from *B. megaterium* ATCC 13368 | *E. coli* BL21-Gold(DE3) | This study |
| pET106A | pET106 derivative with *alkL* gene from *P. putida* GPo1 | *E. coli* BL21-Gold(DE3) | This study |

For detailed information about plasmid construction, see S1 Table. Required genes were either amplified from existing plasmids, genomic DNA, or ordered as gene synthesis constructs. Enzymes (Phusion High Fidelity Polymerase, Q5 High Fidelity Polymerase, restriction enzymes, T5 exonuclease, Taq DNA ligase), dNTPS, and the corresponding buffers were obtained from Thermo Scientific Molecular Biology (St. Leon-Rot, Germany) or New England Biolabs (Frankfurt/Main, Germany). Plasmid isolation and purification of DNA from agarose gels or PCR mixtures were conducted using respective kits from Macherey-Nagel (Düren, Germany) according to supplier protocols. The Gibson Master Mix was prepared according to Gibson et al. [47]. Successful cloning was confirmed by sequencing (Genewiz Germany GmbH, Leipzig, Germany).

To construct the plasmid pETM11-ksa14mutations-alkL, the gene *ksa14mutations* was ordered as gene synthesis construct from Eurofins. The vector pETM11-ksa14m-alkL [17] was used as backbone. Both *ksa14mutations* and the pETM11-ksa14m-alkL were digested with *Nco*I and *Sac*I, followed by ligation with T4 DNA Ligase (Thermo Fisher) according to the manufacturer's instructions.

## Site-directed mutagenesis of *ksa14m*

Site-directed mutagenesis PCR was performed in order to incorporate the desired nucleotide exchanges into the *ksa14m* gene (for details see S2 Table). Desired plasmids were pETM11-ksa14m(M82A)-alkL and pETM11-ksa14m(L188P)-alkL, respectively. The forward primer contained the target mutation and included at least 10 complementary nucleotides on the 3' side of the mutation. The reverse primer was designed so that the 5' ends of the two primers anneal back-to-back. The vector pETM11-ksa14m-alkL [17] served as DNA template. PCR resulted in the formation of non-nicked, linearized plasmids. DNA was purified from the reaction mixtures using the NucleoSpin® Gel and PCR Clean-up kit (Macherey-Nagel) and then digested with *Dpn*I to remove methylated template DNA. The resulting reaction mixture was introduced into *E. coli* DH5α and desired mutations were verified *via* sequencing (Genewiz Germany GmbH).

## Cultivation of microbial strains

Cell cultivations were carried out in baffled Erlenmeyer shake flasks with a liquid volume of maximally 20% of the total volume in a Multitron shaker (Infors, Bottmingen, Switzerland). Microorganisms were either grown in lysogeny broth medium (LB) [40] or in minimal media (M9 [40], M9* [35], or Riesenberg (RB) [48]) with a pH of 7.2 (see S3 Table for media compositions and preparation). Minimal media contained 0.5% (w/v) D-glucose as sole carbon and energy source. Kanamycin (50 µg mL$^{-1}$) or chloramphenicol (34 µg mL$^{-1}$) were added when appropriate.

*E. coli* cells from a frozen glycerol stock were cultivated in LB medium at 37°C and 200 rpm for 6-8 h. This preculture was used to inoculate an M9, M9*, or RB preculture (1% v/v). Incubation was continued at 30°C and 200 rpm for 14–16 h. From this preculture, M9, M9*, or RB main cultures were inoculated to an optical density of 0.2 at 450 nm (OD$_{450}$). Heterologous gene expression was induced in the early exponential phase (OD$_{450}$ ~0.6) by addition of isopropyl β-D-1-thiogalactopyranoside (IPTG) for P$_{T7}$/P$_{lacUV5}$/P$_{tac}$-based vectors or dicyclopropyl ketone (DCPK) for the P$_{alkB}$-based regulatory system (concentrations are indicated in the figure legends and corresponding descriptions). Simultaneously, 0.5 mM of the heme precursor 5-aminolevulinic acid were added for enhanced heme synthesis. Incubation was continued either at 30°C for 5 h or at 20°C for 16 h. Cells were then harvested by centrifugation (5,000 g,

5 min) at 4 or 20°C to be employed in biotransformation experiments with resting and growing cells, respectively.

Cultivation of *P. taiwanensis* was similar to that of *E. coli* with the following adjustments: LB precultures were incubated at 30°C and 200 rpm for 20 h. M9 pre- and main cultures were treated as described for *E. coli* except for the addition of 5-aminolevulinic acid. Streptomycin (100 μg mL$^{-1}$) was added to LB pre-cultures, but not in minimal media. Cells were harvested 5 h after induction by centrifugation (5,000 g, 5 min) at 20°C for resting-cell biotransformations [45].

## Biotransformation with resting *E. coli* cells

After cultivation, induction, and harvesting as described above, resulting cell pellets were washed once and then resuspended in different buffers to a cell concentration of 1 g$_{CDW}$ L$^{-1}$ (unless stated otherwise). Buffers included 100 mM potassium phosphate buffer (pH 7.4, KP$_i$ buffer) supplemented with 1% (w/v) glucose for a resting-cell state based on nitrogen limitation and M9 medium (pH 7.4) lacking magnesium and containing 1% (w/v) glucose for a resting-cell state based on magnesium limitation. Resting-cell biotransformations were performed at 30°C and 250 rpm at a 1 mL-scale in screw-capped glass tubes (12 mL) and were initiated after 15 min of equilibration by addition of 10 μL of a 100 mM steroid stock solution in DMSO, resulting in final concentrations of 1 mM steroid and 1% (v/v) DMSO. Testosterone was used as substrate for KSA14m and CYP106A2, whereas progesterone was added for CYP154C5-catalyzed steroid hydroxylation. One hundred μL sample were taken after different time intervals and mixed with 12.5 μL HCl (1 M) to stop the reaction. Pure acetonitrile was added (50% v/v) to dissolve precipitated steroids, followed by mixing on a ThermoMixer C (Eppendorf, Hamburg, Germany at 2000 rpm, 5 min, 4°C) and centrifugation (17,000 g, 5 min, 4°C) for biomass removal. The resulting supernatant was stored at -20°C until further analysis. All assays were conducted in biological duplicates for each condition. Specific activities are given in U g$_{CDW}$$^{-1}$, with 1 U defined as the activity forming 1 μmol of product per min, and were retrieved *via* quantification of accumulated product divided by the biomass concentration applied.

## Biotransformation with growing *E. coli* cells

Cells were harvested before and 5 h after induction. Resulting pellets were washed once and resuspended in pre-warmed M9 medium (S3 Table) including IPTG, 5-aminolevulinic acid, and 1% (w/v) glucose. Ten mL of sample were filled into screw-capped, baffled flasks (100 mL) and equilibrated at 30°C and 250 rpm for 15 min. Biotransformations were started by adding 100 μL of a 100 mM testosterone stock solution in DMSO, leading to final concentrations of 1 mM testosterone and 1% (v/v) DMSO. Reaction termination and sampling were conducted as described above for resting-cell biotransformations.

## Biotransformation with resting *P. taiwanensis* cells

Fresh cells were harvested from M9 cultures 5 h after induction and applied in nitrogen-limited resting-cell activity assays as described above for *E. coli*. Cell harvesting and resting-cell preparation were conducted at room temperature.

## Analytical methods

Biomass concentrations were determined photometrically as the optical density at a wavelength of 450 nm (Libra S11 spectrophotometer, Biochrom Ltd., Cambridge, UK). One OD$_{450}$

unit corresponds to 0.166 $g_{CDW}$ $L^{-1}$ for *E. coli* [49] and 0.186 $g_{CDW}$ $L^{-1}$ for *P. taiwanensis* [50], respectively.

Monitoring of protein synthesis was performed by harvesting 80 μg of cell dry weight (CDW) from cultures for sodium dodecyl sulfate-polyacrylamide gel electrophoresis (SDS-PAGE) according to Laemmli [51]. Proteins extracted from 15 $μg_{CDW}$ were loaded per lane and stained with Coomassie Brillant Blue R-250. PageRuler™ Prestained Protein Ladder (Thermo Fisher Scientific, Waltham, MA, USA) was used as reference.

Testosterone and respective product (2β- and 15β-hydroxytestosterone) concentrations were determined *via* HPLC using a Dionex Ultimate 3000 system (Thermo Fisher Scientific) equipped with a Syncronis C18 column (150 x 2.1 mm, 3 μm particle size, Thermo Fisher Scientific) and an UV detector operating at 245 nm for steroid detection. After injection of 5 μL sample, steroids were eluted at a column oven temperature of 40˚C with 55% acetonitrile in ultrapure water as a mobile phase at a flow rate of 0.5 mL $min^{-1}$. Quantification was based on peak areas and calibration curves prepared with commercially available standards.

Progesterone and 16α-hydroxyprogesterone were quantified using the same HPLC system equipped with an Accucore C18 column (150 x 3 mm, 2.6 μm particle size, Thermo Fisher Scientific). An eluent consisting of 50% acetonitrile in ultrapure water was used at a flow rate of 0.6 mL $min^{-1}$. Column oven temperature, sample volume, detection, and quantification were as described above.

## Results and discussion

In our previous work, *E. coli* BL21-Gold(DE3)_pETM11-ksa14m-alkL carrying the BM3 variant KSA14m and the hydrophobic outer membrane pore AlkL has been developed (S1A Fig), enabling high-rate testosterone hydroxylation to 15β-hydroxytestosterone as main product (85%) [17]. Poor operational stability, however, was observed for this biocatalyst in the nitrogen-deprived resting-cell state [22]. In this study, we investigate various strategies to counteract whole-cell biocatalyst destabilization with the ultimate goal to enable high process durability and thus efficiency.

### Relieving nitrogen limitation does not stabilize testosterone hydroxylation rates

Approaches based on resting (i.e., metabolically active non-growing) cells decouple growth and biocatalyst synthesis from the production phase. Their application allows growth-independent investigation and optimization of desirable reactions, can minimize side reactions, and enables the identification of potential limitations of the overall performance [18, 30, 52]. Compared to growing cells, resting cells have proven beneficial for redox biotransformations with respect to specific activities [29, 30, 33, 53, 54] as well as reusability [55]. However, establishing the resting-cell state represents a transition from the availability of all essential nutrients to a limitation of one or more nutrients [31]. Beside respective technical efforts, this involves changes in microbial physiology including intracellular metabolite concentrations and metabolic fluxes [56], which depend on the type of limitation and ultimately affect heterologous protein synthesis. For example, nitrogen-deprived *E. coli* cells displayed lower glucose uptake as well as pyruvate and acetate secretion rates than magnesium-limited cells [33, 52]. Consequently, the type of nutrient limitation is decisive for resting-cell phenotypes. Omitting nitrogen from the biotransformation medium is prone to effect a limited amino acid and protein synthesis capacity [57], putatively leading to poor stability of (heterologous) enzyme levels/activities under production conditions as observed for the investigated testosterone-hydroxylating whole-cell biocatalyst. Furthermore, Fe(III) constitutes an essential component

of the heme center located in the active site of CYP450s and may become limiting [7]. Interestingly, employing resting cells in a complete growth medium lacking only magnesium has been reported to stabilize limonene formation by engineered *E. coli* compared to their application in nitrogen-deprived KP$_i$ buffer [33]. Thereby, the application of complete M9 medium lacking only nitrogen did not result in differences compared to KP$_i$ buffer with glucose. Applying M9 medium just lacking magnesium did, however, not significantly improve nor stabilize specific KSA14m-based testosterone hydroxylation rates, which still dropped by 30-36% within the first 30 min of the reaction (Fig 1A). Also inducer addition to resting cell media did not lead to a stabilization, whereas this has been reported to partially relieve enzyme instability issues in previous studies [28, 33]. Relative KSA14m levels were stable for 1 h but not detectable after 24 h of biotransformation (Fig 1B). This is in stark contrast to AlkL, for which the SDS-PAGE band even intensified after 24 h with the inducer present under magnesium-limited conditions. The BM3 variant obviously decayed under biotransformation conditions, although gene expression from the same operon still occurred to a considerable extent. Altogether, the poor KSA14m-based biocatalyst stability could not be alleviated by facilitating protein re-synthesis *via* nitrogen supply and magnesium limitation or continuous induction. Generally, biocatalytic activities of resting recombinant cells often have been observed to decrease steadily, e.g., due to limited enzyme stability and resynthesis [33, 58–61]. CYP450 resynthesis may become limiting when biosynthetic and thus enzyme synthesis capacities are down-regulated in a non-growing state, gene expression profiles are altered [62], and cellular resource demands increase due to toxic effects of enzymes and product [30]. Therefore, cells growing in media containing all necessary components for (re-)synthesis often convey superior stability, possibly favoring this biocatalyst configuration for process setups.

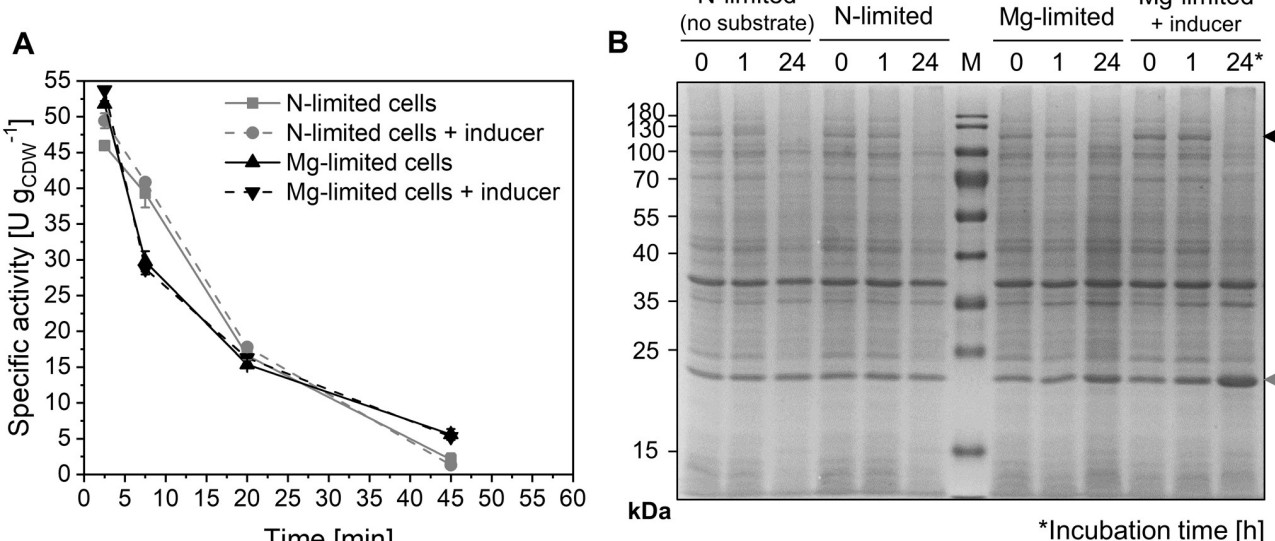

**Fig 1. Biotransformations with resting *E. coli* BL21-Gold(DE3)_pETM11-ksa14m-alkL cells limited in either nitrogen (N) or magnesium (Mg).** Microorganisms were grown in M9 medium containing 0.5% (w/v) glucose. Resting cells were prepared 5 h after induction, and biotransformations were performed as described in Materials and Methods. (A) Specific testosterone hydroxylation activity time courses for cells suspended in either N-free KP$_i$ buffer or Mg-free M9 medium, optionally supplied with 0.1 mM IPTG and 0.5 mM 5-aminolevulinic acid (inducer, dashed lines). Data points represent average values and standard deviations of biological duplicates. (B) SDS-PAGE analysis shows KSA14m (119 kDa, black arrow) and AlkL (23 kDa, grey arrow) levels during biotransformations under the indicated conditions: N-deprived cells in absence or presence of substrate (testosterone), Mg-limited cells with substrate in absence or presence of IPTG and 5-aminolevulinic acid (inducer and heme building block).

### Growing-cell biotransformation allows continuous KSA14m synthesis, but does not lead to activity stabilization

In contrast to resting cells, growing cells enable efficient (re-)synthesis of cells, enzymes, and energy/redox carriers [18, 63]. This biocatalyst format thus is considered favorable for the handling of instable enzymes as well as biocatalysis-related stress, such as substrate- and/or product-related toxicity, inhibition, or oxidative stress [30, 34, 64]. As this becomes especially important for catalysis by instable CYP450s, we studied the catalytic performance of growing *E. coli* BL21-Gold(DE3)_pETM11-ksa14m-alkL cells (Fig 2).

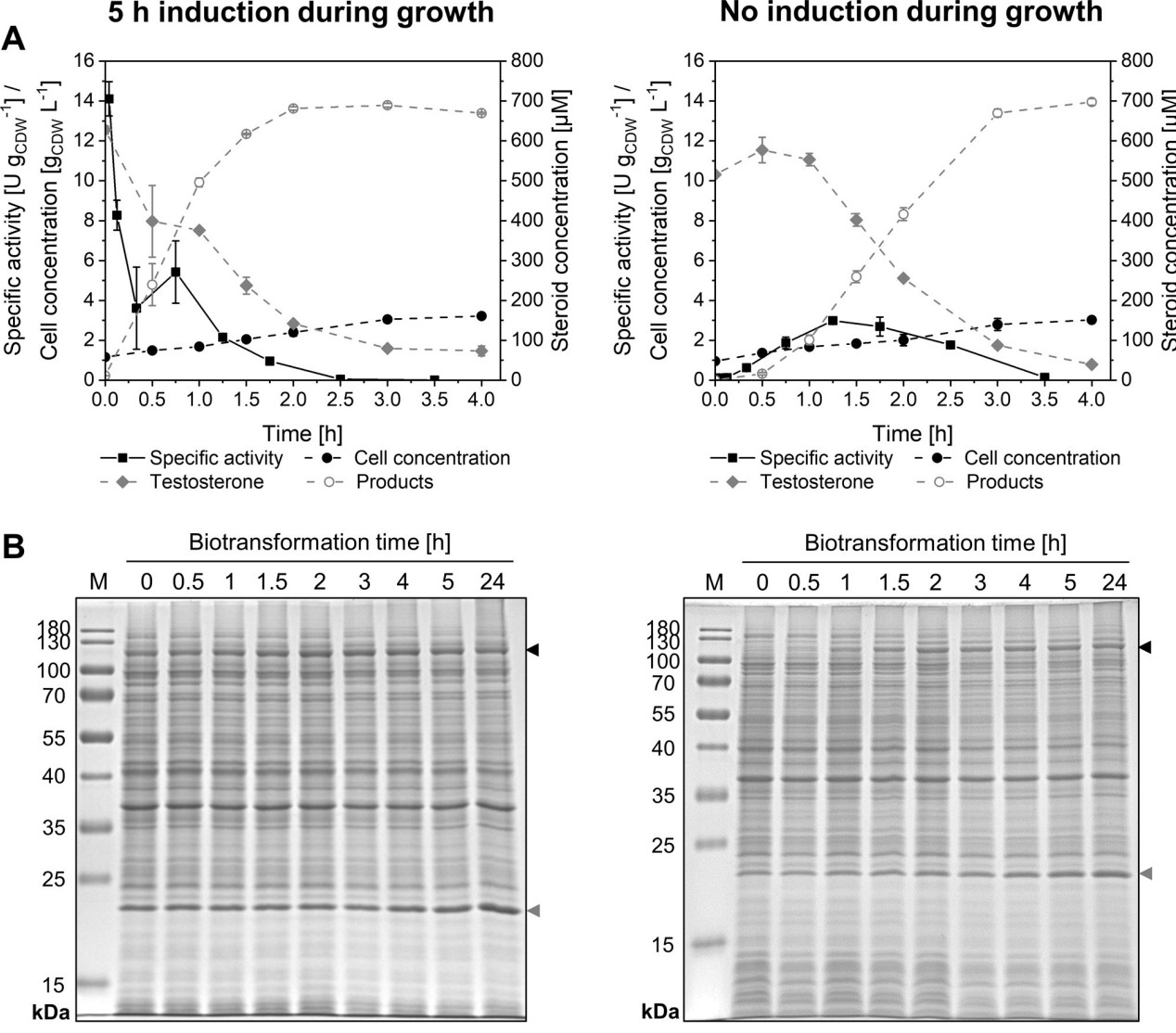

**Fig 2. Biotransformations with growing *E. coli* BL21-Gold(DE3)_pETM11-ksa14m-alkL cells.** Bacteria were cultivated in M9 medium with 0.5% (w/v) glucose, induced with 0.1 mM IPTG, harvested before and after 5 h of induction, and resuspended in fresh M9 medium containing 1% (w/v) glucose, 0.1 mM IPTG, and 5-aminolevulinic acid to a biomass concentration of 1 $g_{CDW}$ $L^{-1}$. Ten mL liquid volume were transferred to 100 mL baffled, screw-capped flasks and equilibrated for 10 min at 30°C. Biotransformations were started by addition of 1 mM testosterone. (A) Time courses of specific testosterone hydroxylation activities (solid black lines), cell concentrations (dashed black lines), and steroid concentrations (dashed grey lines; products referring to the sum of 2β- and 15β-hydroxytestosterone) with 5 h of induction before and induction simultaneously to the biotransformation start. Average values and standard deviations were calculated from two biological replicates. (B) SDS-PAGE analyses showing heterologous KSA14m (119 kDa, black arrows) and AlkL (23 kDa, grey arrows) levels at different time points after biotransformation start.

Biotransformations initialized after 5 h of induction displayed initial specific product formation rates of $14.1 \pm 0.9$ U $g_{CDW}^{-1}$ (Fig 2A, left), which accounted for only 31% of rates obtained with resting cells prepared after the same induction time ($46.0 \pm 0.1$ U $g_{CDW}^{-1}$, Fig 1A). Similar findings have been reported for recombinant *E. coli* capable of styrene epoxidation [30], cyclohexanone Baeyer-Villiger oxidation [54], naphthalene oxidation [53], or *P. taiwanensis* harboring an enzyme cascade producing 6-hydroxyhexanoic acid from cyclohexane [28]. Lower growing-cell activities typically result from a stronger competition of enzyme synthesis and metabolic demands of the reaction with demands for cellular maintenance and biomass formation [18, 29]. However, opposite observations exist for styrene-epoxidizing *P. taiwanensis* VLB120 strains [65–67]. In contrast to resting-cell biotransformations, KSA14m levels were however stable for 24 h of biotransformation (Fig 2B, left), confirming continuous enzyme synthesis by growing cells. Despite stable KSA14m levels and available testosterone, specific activities declined rapidly (Fig 2A, left). Product formation was detected for 2 h of biotransformation, which was not significantly longer compared to resting cells [22]. For cells induced simultaneously with substrate addition, specific activities slowly increased (Fig 2A, right) correlating with increasing KSA14m levels (Fig 2B, right). A maximum of $3.0 \pm 0.1$ U $g_{CDW}^{-1}$ was reached after 1.5 h, followed by an activity decrease despite stable KSA14m levels and sufficient testosterone availability for at least 2 h of biotransformation. The weak product inhibition of the system [22] likely can be excluded as the cause for the activity decrease, which may rather be caused by the loss of heme and/or FMN/FAD as the prosthetic groups of the oxygenase and/or reductase domains of CYP450 BM3, respectively. These aspects remain to be evaluated in future studies. Obviously, KSA14m experienced a similar inactivation in growing as in resting cells, which could not be alleviated by enzyme resynthesis in growing cells. After 5 h of induction and with the same cell and initial testosterone concentrations, resting cells reached clearly higher activities leading to faster initial product accumulation (S2 Fig). The higher product titer reached within 1 h correspond to a superior space-time yield of 0.25 vs. 0.15 g $L^{-1}$ $h^{-1}$. Resting cells therefore remained the format of choice for further studies.

## Cultivation medium and expression strategies influence initial whole-cell activity, but not its stability

Beside product inhibition, we previously proposed uncoupling and ROS formation as a critical factor possibly affecting biocatalyst stability [22] either in terms of cell toxification or enzyme deactivation [24–26]. The latter may manifest in active site destruction, as it has been reported for the heme cofactor in CYP450s [68, 69]. We first investigated effects of medium composition and differential *ksa14m* expression, in order to foster functional enzyme synthesis.

As previously demonstrated for a 6-hydroxyhexanoic acid-producing *P. taiwanensis*, medium composition is a core factor influencing cell physiology and catalytic performance [28]. Effects of cultivation in different standard minimal media, i.e., M9, M9*, and RB [40, 48], on *E. coli* BL21-Gold(DE3)_pETM11-ksa14m-alkL were investigated in terms of specific growth rate, heterologous protein synthesis, and specific testosterone hydroxylation activities. Compared to M9, M9* contains the three-fold concentration of phosphate salts resulting in higher buffering capacity, whereas RB is characterized by increased ammonium concentrations and intermediate phosphate concentrations (S3 Table).

The specific growth rate did not considerably differ in M9 and RB media and was 18% lower in M9* medium (Fig 3A), while induction in M9* medium led to higher relative KSA14m levels compared to M9 and RB media (Fig 3B). Initial testosterone hydroxylation activities obtained in the resting cell format were however highest after growth in M9 medium, being 24 and 45% lower after growth in RB and M9* medium, respectively (Fig 3C). Thereby,

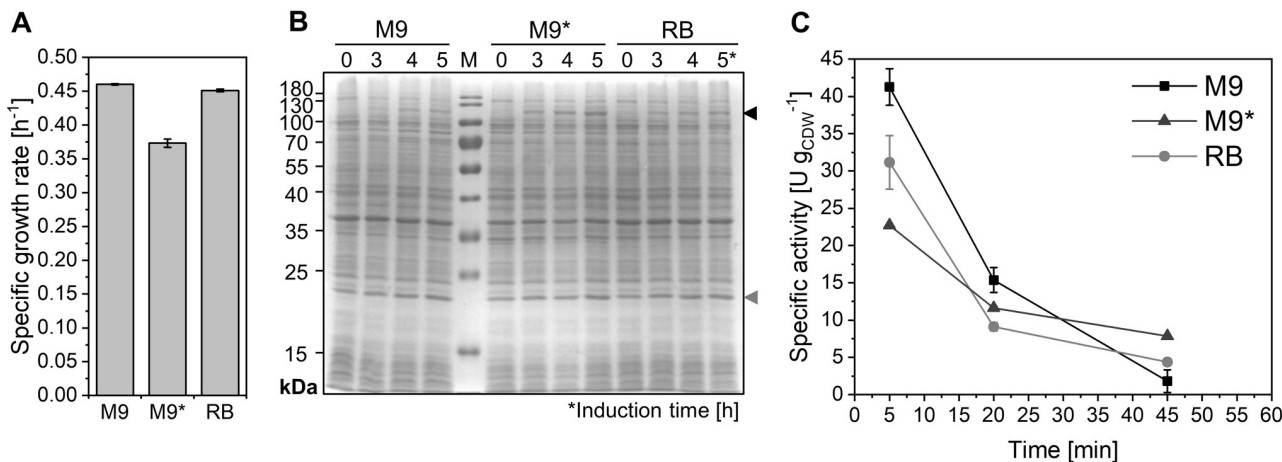

**Fig 3. Performance of *E. coli* BL21-Gold(DE3)_pETM11-ksa14m-alkL depends on cultivation medium composition.** Cells grown and induced in M9, M9*, or RB medium containing 0.5% (w/v) glucose as carbon and energy source. Specific growth rates (A), KSA14m (119 kDa, black arrow) and AlkL (23 kDa, grey arrow) levels (SDS-PAGE, B) and resting-cell activity courses after 5 h of induction in nitrogen-deprived KP$_i$ buffer containing 1% (w/v) glucose (C) were analyzed as described in Materials and Methods. Average values and standard deviations of biological duplicates are shown.

variations in cell physiology depending on the growth medium applied may play a role, possibly involving differences in metabolic by-product accumulation and respective inhibitory effects on bacterial growth [70, 71]. Additionally, changed metabolic fluxes may influence the availability of resources for (heterologous) enzyme synthesis and catalysis, which in turn can result in differing amounts of functional enzyme, even when the overall levels of a specific enzyme remain similar. Altogether, the biocatalyst performance obtained upon cultivation in M9 medium exceeded that achieved with other minimal media. In all cases, however, activities decreased rapidly during resting-cell biotransformations. These observations again indicate that high oxygenase (KSA14m) levels do not necessarily result in high activities. This also has been observed in other studies and was connected to negative effects on the host cell physiology, which could be overcome by fine-tuning of oxygenase levels in the cell [17, 29, 72, 73].

The construct pETM11-ksa14m-alkL used so far features the *lacI*-P$_{T7}$-based regulatory system (S1A Fig) allowing fast and strong expression of *ksa14m* and *alkL*. Beside differences in the molecular environment in host strains compared to the wildtype strain possibly affecting functional expression and catalysis [11], plasmid maintenance and heterologous gene (over) expression may impose a metabolic burden [27]. Other critical factors complicating stable and functional CYP450 synthesis include protein folding and heme incorporation. Further, cofactor regeneration *via* the cellular metabolism may not keep up with the high catalytic rates enabled by the enzyme [74], and high oxygenase activities may lead to enforced ROS formation, which can affect both host and oxygenase stability [24, 73, 75]. Taken together, fine-tuning of *ksa14m* expression on the genetic level appears promising to stabilize testosterone hydroxylation by recombinant *E. coli* [36]. Recently, a doubling of CYP450-mediated cyclohexane hydroxylation activity has been achieved by regulatory system engineering in recombinant *P. taiwanensis* VLB120 [72]. We thus tested different expression systems and inducer concentrations to vary gene expression rates and levels and investigate respective effects on whole-cell biocatalyst activity and stability. New vectors were based on the pCOM10 plasmid backbone carrying regulatory systems based on IPTG-inducible *lacI*-P$_{lacUV5}$ or *lacI$^q$*-P$_{tac}$ or DCPK-inducible *alkS*-P$_{alkB}$ (S1B–S1D Fig) [44–46]. These all have in common a lower promoter strength than the so far applied P$_{T7}$ promoter (P$_{T7}$ > P$_{tac}$ ≈ P$_{alkB}$ > P$_{lacUV5}$) and are

considered as easier titratable by varying the concentration of respective inducers [76–79]. These characteristics allow the investigation of whole-cell biocatalyst performance upon variation of transcription and ultimately expression levels. The resulting constructs plac-ksa14m-alkL, ptac-ksa14m-alkL, and palk-ksa14m-alkL were introduced and tested in *E. coli* BL21-Gold(DE3).

Applying inducer levels known to effect varying induction strength, the highest initial activities were achieved with low inducer concentrations with all constructs whereas stronger induction led to a 34–79% lower initial activity depending on the construct applied (Fig 4). At low inducer concentrations, the constructs ptac-ksa14m-alkL and pETM11-ksa14m-alkL allowed the fastest growth ($\mu$ = 0.42 h$^{-1}$) and the highest initial activities (32.6 ± 0.2 and 45.2 ± 0.9 U $g_{CDW}^{-1}$, respectively). The hampered and non-exponential growth observed for most recombinant strains at elevated inducer concentrations indicated a significant metabolic burden, likely imposed by high gene expression rates competing with energy demands for growth [27]. In the case of plac-ksa14m-alkL, growth also was impaired at low IPTG concentrations, possibly due to metabolic burden related to the maintenance of this particular plasmid [80]. Interestingly, relative KSA14m levels remained unaffected by inducer titration (S3A, S3B and S3D Fig) except for ptac-ksa14m-alkL, with which higher IPTG concentrations led to higher relative oxygenase levels (S3C Fig). The mere increase in enzyme amount, however, was not associated with higher whole-cell activities (Fig 4C). Especially upon strong induction, a large CYP450 fraction likely was not appropriately processed in terms of folding and/or heme incorporation. Activity further may have been limited by NADPH supply *via* the stressed cell metabolism as well as uncoupling-related effects on cell physiology and enzyme performance [27]. Altered expression systems and inducer concentrations may also have affected active AlkL levels provoking positive or negative effects on whole-cell biocatalyst performance in terms of improved substrate availability or toxic intracellular substrate and/or product titers, respectively [17, 81–83].

However, specific activities were instable for all tested strains independently of inducer concentration and expression system used (Fig 4, right column). As the activity drop for whole-cell biocatalysts with high initial activity can be due to fast testosterone depletion [22], the effect of substrate availability was further investigated by reducing the biocatalyst concentration applying the same initial testosterone concentration (1 mM), which ensured an increased availability of both testosterone and $O_2$. Initial activities were the same, excluding $O_2$ limitation as a critical factor (Fig 4B and 4C). For the strain carrying ptac-ksa14m-alkL, the instability was largely preserved (Fig 4C). Interestingly, the strain carrying plac-ksa14m-alkL exhibited a higher stability with 10-fold reduced cell concentration (Fig 4B, right column). A similar, but less pronounced stabilization of testosterone hydroxylation rates upon biocatalyst concentration decrease also has been observed for *E. coli* carrying pETM11-ksa14m-alkL [22], which however showed significantly higher initial activities (Fig 4B and 4D). In this respect, the mentioned weak product inhibition and limited testosterone availability have to be considered [22], being less prominent at low biocatalyst concentration and thus product accumulation.

In summary, the tested variations of expression system and inducer concentrations did not result in significantly improved initial biocatalyst activity or stability (except for plac-ksa14m-alkL), but showed that the construct pETM11-ksa14m-alkL enables comparatively efficient expression along with fast growth. Further attempts to stabilize the activity *via* genetic engineering should follow a holistic and systematic engineering approach involving the dissection of effects related to transcription (*via* promoter), translation (*via* ribosomal binding site), and gene dosage (*via* origin of replication), as it has been performed for CYP450-catalyzed cyclohexane hydroxylation [72].

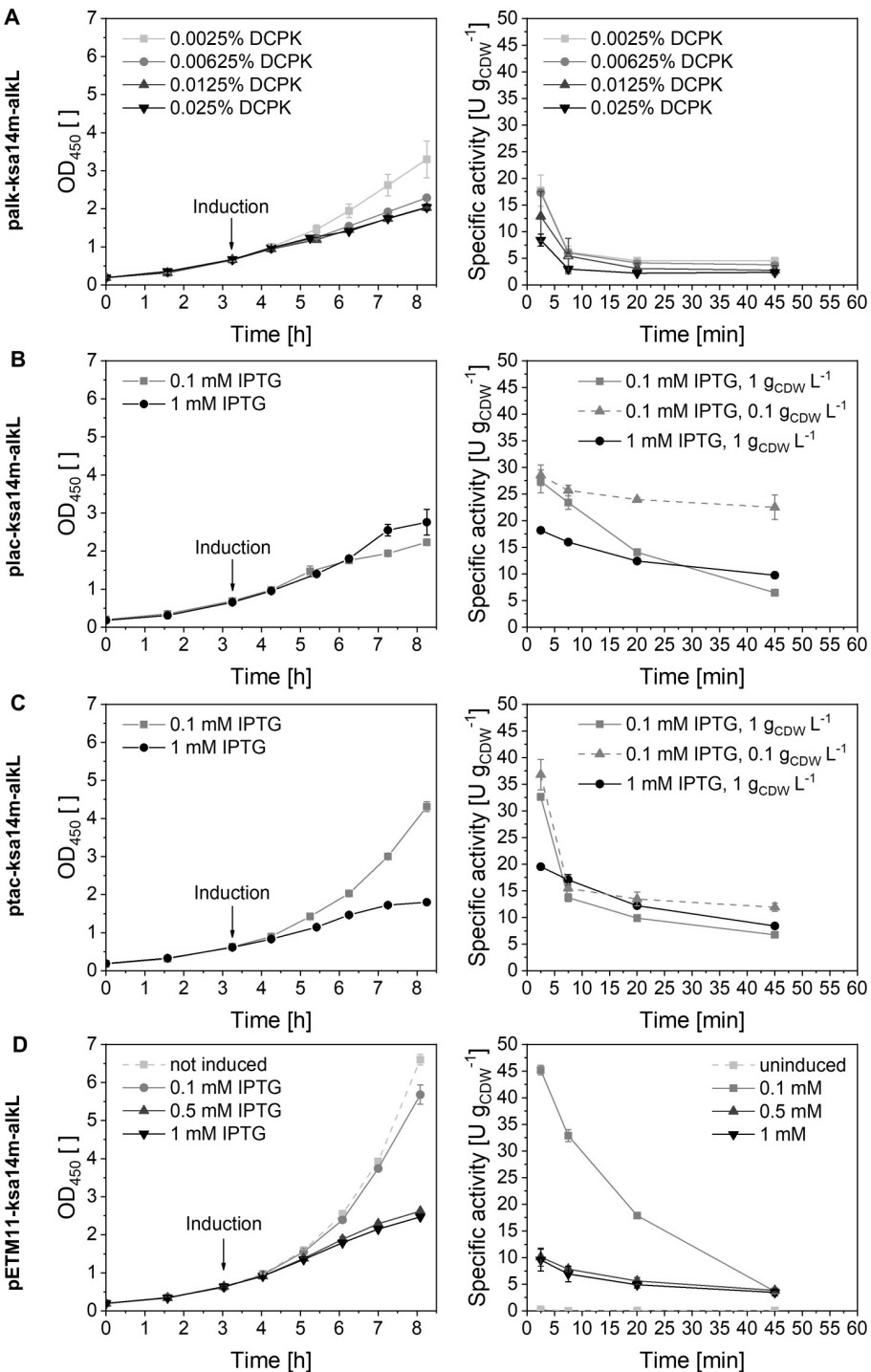

**Fig 4. Effects of expression system and inducer concentration on the performance of *E. coli* BL21-Gold(DE3) strains synthesizing KSA14m and AlkL.** Left and right columns show courses of growth and specific resting-cell testosterone hydroxylation activities, respectively, upon *ksa14m* and *alkL* expression under control of different regulatory systems (S1 Fig) and with different inducer concentrations. (A) AlkS-P$_{alkB}$-based system induced with DCPK; (B) LacI-P$_{lacUV5}$-based system induced with IPTG; (C) LacI$^q$-P$_{tac}$-based system induced with IPTG; (D) LacI-P$_{T7}$-based system either uninduced or induced with IPTG. Cells were grown in M9 medium containing 0.5% (w/v) glucose and induced for 5 h, followed by cell harvesting and resting-cell biotransformations performed as described in Materials and Methods in nitrogen-deprived KP$_i$ buffer containing 1% (w/v) glucose with a cell concentration of 1 g$_{CDW}$ L$^{-1}$ unless indicated otherwise. Average values and standard deviations of two biological replicates are given.

## Protein engineering with focus on uncoupling

Protein engineering has been reported to be a valuable option to increase turnover numbers and (thermo-)stability of CYP450s, including BM3 [38, 84]. Various enzyme properties such as catalytic rate, specificity, and stability can be addressed either by rational design or directed evolution. Testosterone-hydroxylating BM3 variants were generated by engineering the key active site residue F87, i.e., by introducing the smaller amino acid alanine, thereby vacating space for the bulky non-native substrate testosterone [16]. However, a typical side effect of this approach is uncoupling, occurring upon suboptimal positioning of the substrate and/or binding of the product molecule in the active site [85–87]. Compared to wildtype BM3, converting natural substrates highly coupled to NADPH oxidation (93–96% for long-chain fatty acids) [88, 89], the single mutant F87A indeed exhibits a poor coupling efficiency of 6.5% for testosterone hydroxylation [16]. Uncoupling-related effects on whole-cell biocatalyst efficiency include an increased $O_2$ and NADPH demand and ROS-related enzyme/cell inactivation/destabilization [4, 5, 18, 73, 90]. Mutations improving the substrate fit in the active site have proven successful to reduce uncoupling, e.g., for 2-hydroxy biphenyl 3-monooxygenase [91] or CYP101 [92]. Also the coupling efficiency of BM3 variants has been improved, with the optimization of propane-hydroxylating variants as prime example ($P450_{PMO}$) [93].

Despite its tendency towards uncoupling after amino acid substitution, further mutation of F87A was not feasible as it enables testosterone conversion in the first place. Alternatively, mutation of position 82 has been reported to foster uncoupling [94], which is why we evaluated KSA14m variants with the original alanine being reintroduced at that position. For $P450_{PMO}$, further mutations were found to enhance coupling efficiency, whereby the single mutation L188P caused the largest effect [37]. Based on these promising results, we constructed three new KSA14m variants that either carried single mutations, M82A (KSA14m_M82A) or L188P (KSA14m_L188P), or were equipped with stabilizing mutations of $P450_{PMO}$ (L52I, A184V, I366V, G443A and D698G = KSA14mutations) [37].

Growth of *E. coli* BL21-Gold(DE3) strains harboring respective variants and AlkL was not affected by these mutations (S4 Fig). During resting-cell biotransformations of testosterone, strains with KSA14mutations displayed exactly the same activity levels and instability as those carrying KSA14m (Fig 5A). Interestingly, the single mutation M82A caused a 34% reduced initial activity, indicating its significant role in testosterone hydroxylation, while L188P did not influence initial rates. M82A did not stabilize activities, whereas L188P appeared to cause a slightly slower activity decrease (Fig 5A). To investigate biotransformation-independent whole-cell biocatalyst stability, the normally applied equilibration time under reaction conditions prior to biotransformation start (0.25 h) was prolonged. Initial specific testosterone hydroxylation activities of all BM3 variants declined with extended pre-incubation, revealing a rather high biotransformation-independent instability (Fig 5B) as previously reported for the original whole-cell biocatalyst harboring KSA14m [22]. In contrast to the activity drop during biotransformation, the fastest decrease was observed for the L188P variant.

In conclusion, the chosen mutations did not significantly stabilize whole-cell testosterone hydroxylation activities. It has to be considered that most mutations tested improved coupling of a BM3 variant optimized for the substrate propane [37], which obviously differs from testosterone in terms of bulkiness and polarity. Future engineering attempts thus may focus on improved testosterone binding, as low $K_M$ values have been found to correlate with high coupling efficiencies [93], and benefit from the recent development of computational methods capable of three-dimensional protein structure prediction from amino acid sequences, such as AlphaFold [95].

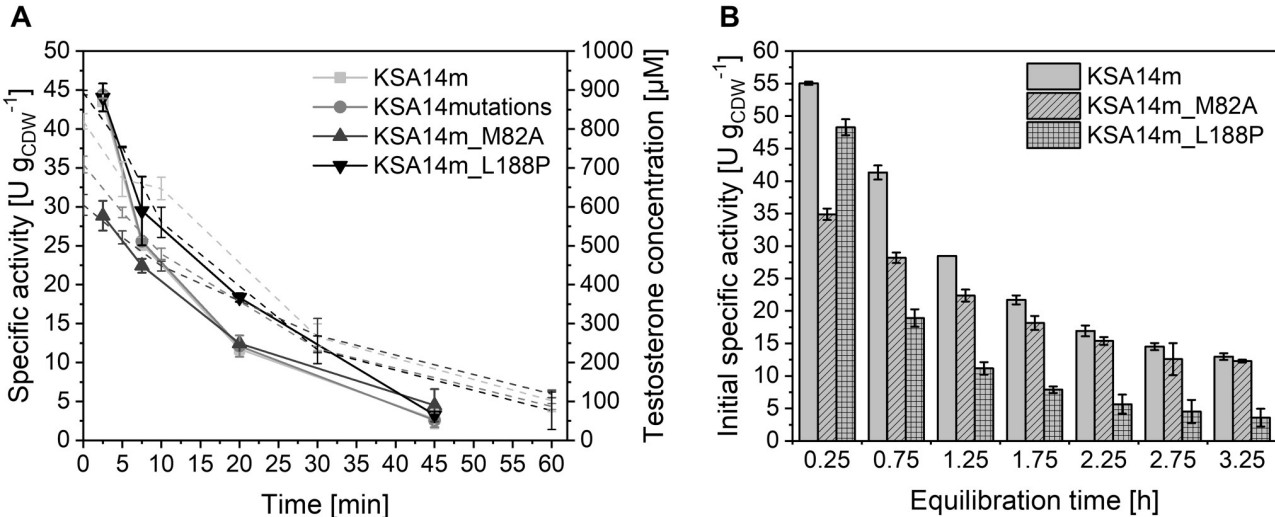

**Fig 5. Effect of amino acid exchanges in KSA14m on testosterone hydroxylation activity and stability.** *E. coli* BL21-Gold(DE3) cells carrying pETM11 equipped with genes encoding different KSA14m variants and AlkL were grown in M9 medium containing 0.5% (w/v) glucose. Resting cell preparation 5 h after induction with 0.1 mM IPTG and biotransformations were performed as described in Materials and Methods in nitrogen-deprived KP$_i$ buffer containing 1% (w/v) glucose. (A) Time courses of specific testosterone hydroxylation activities (solid lines) and testosterone concentrations (dashed lines). (B) Initial activities (5 min) after different equilibration times under reaction conditions. Average values and standard deviations of biological duplicates are given.

## *P. taiwanensis* as host organism for CYP450 BM3 catalysis

Intrinsic features of the heterologous host organism can crucially impact the functional synthesis and stability of recombinant enzymes. *E. coli* lacks intrinsic CYP450 genes, which minimizes cross reactivities. However, despite the existence of hemoproteins like cytochrome bc, heme synthesis in *E. coli* is not very efficient, which can hamper active site buildup in KSA14m. In contrast, pseudomonads possess the intrinsic ability to efficiently synthesize heme, which makes the addition of a heme precursor (such as 5-aminolevulinic acid) unnecessary. The strain *P. taiwanensis* VLB120 represents a highly interesting host strain as it can tolerate high solvent levels and provides a high metabolic capacity to support oxygenase catalysis also at high cell densities and also for CYP450s [65, 67, 96, 97]. KSA14m-catalyzed testosterone hydroxylation was thus investigated using *P. taiwanensis* VLB120_Strep. To this end, pCOM10 vectors proven suitable for heterologous oxygenase synthesis in various *E. coli* [44, 98] as well as *Pseudomonas* strains [45, 46] and equipped with *ksa14m* were used. Optionally, the plasmids contained genes encoding hydrophobic outer membrane proteins, for which a facilitated testosterone uptake into *E. coli* has been demonstrated previously [17]. Tested candidates comprised AlkL, TodX (native toluene uptake protein from *Pseudomonas putida* F1 [99]), and FhuA Δ1–160 (truncated importer for ferric hydroxamate from *E. coli* [100]). Growth, protein synthesis, and specific activities were evaluated for *P. taiwanensis* VLB120_Strep strains harboring palk-ksa14m-alkL, plac-ksa14m, plac-ksa14m-alkL, ptac-ksa14m-alkL, plac-ksa14m-fhuAΔ1–160, or plac-ksa14m-todX (Fig 6).

Cells equipped with ptac-ksa14m-alkL did not grow at all in LB precultures. The other plasmid-containing strains grew more slowly than the plasmid-free wild type, even before induction (Fig 6A), indicating leaky gene expression. A second exponential phase started ~1 h after induction. For the palk-ksa14m-alkL strain, KSA14m was not visible on SDS-PAGE gels (Fig 6B), and testosterone hydroxylation was not detected. With all other plasmids (based on

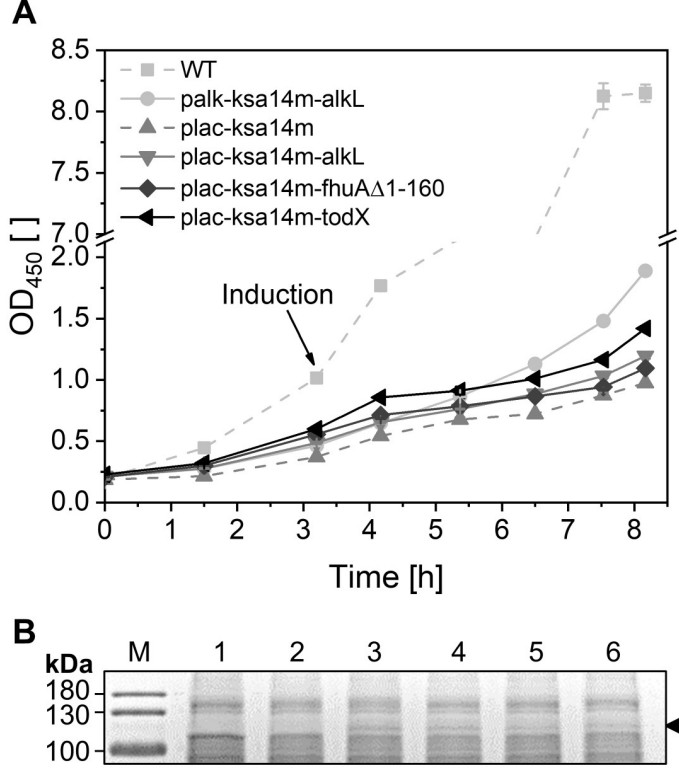

**Fig 6.** *P. taiwanensis* **VLB120_Strep hosting the heterologous synthesis of KSA14m and different hydrophobic outer membrane pores.** (A) Growth of *P. taiwanensis* VLB120_Strep strains equipped with different plasmids for KSA14m and hydrophobic outer membrane pore synthesis in M9 medium containing 0.5% (w/v) glucose compared to the wild type (WT). Data points represent average values and standard deviations of biological duplicates. Heterologous gene expression was induced with 0.0025% (v/v) DCPK or 0.1 mM IPTG. (B) Recombinant KSA14m levels (119 kDa, black arrow) were examined *via* SDS-PAGE 5 h after induction for the wild type (1) and strains harboring either palk-ksa14m-alkL (2), plac-ksa14m (3), plac-ksa14m-alkL (4), plac-ksa14m-fhuAΔ1-160 (5), or plac-ksa14m-todX (6).

pCOM10_lac), low KSA14m levels (Fig 6B) and low specific product formation rates (1.2–2.4 U $g_{CDW}^{-1}$) were observed. The similar activities with and without pores are not surprising since testosterone uptake only becomes limiting with higher intracellular steroid conversion activities. It however remains unclear, if outer membrane proteins were synthesized (not visible on SDS-PAGE gels, see Fig 6B in S1 Raw images). Carrying the same broad host range plasmid plac-ksa14m-alkL, *P. taiwanensis* VLB120_Strep showed 14-fold lower specific activities than *E. coli* BL21-Gold(DE3), rendering the latter by far superior for testosterone hydroxylation. Therefore, employing *E. coli* strains featuring improved heme uptake or production may constitute an interesting option for CYP450 BM3-based steroid hydroxylation [101–103]. This however does not dismiss pseudomonads as hosts for biocatalytic steroid hydroxylations, since the systematic evaluation of other expression vectors may still improve their performance with regard to growth, heterologous protein synthesis, and catalytic activity [72].

## Alternative steroid-hydroxylating CYP450s display higher stability but low activity

Although initial specific steroid conversion rates achieved with engineered BM3 variants proved to be superior to those reported for native steroid-hydroxylating CYP450s [17], the

latter showed better stability [20, 21, 104, 105]. CYP106A2 from *B. megaterium* ATCC13368 (15β-hydroxylation of testosterone) and CYP154C5 from *Nocardia farcinica* IFM10152 (16α-hydroxylation of progesterone) exhibited the most promising activities [20, 21], surpassing those of mammalian CYP450s or other bacterial representatives [17]. In contrast to self-sufficient CYP450 BM3, these oxygenases depend on additional redox proteins such as the compatible putidaredoxin reductase (PDR, encoded by *camA*) and putidaredoxin (Pd, encoded by *camB*). Most oxygenases constitute multi-component systems, which can involve a limitation by the electron transfer to the active site. In contrast, self-sufficient systems experience high electron pressure on the active site [24, 73, 75], which can foster uncoupling reactions and concomitant ROS formation. Thus, we evaluated specific activities of cells containing CYP106A2 or CYP154C5 together with PDR and Pd and investigated respective stabilities.

In the first approach, CYP450 genes were combined with *camA*, *camB*, and optionally *alkL* in one single operon under control of the LacI-$P_{T7}$-based regulatory system on the pETM11 plasmid (S5 Fig, one plasmid strategy). Whereas cloning was not successful with *cyp106a2*, vectors pET154-camAB and pET154-camAB-alkL carrying *cyp154c5* were successfully established in *E. coli* BL21-Gold(DE3). Specific growth rates were not considerably hampered upon induction of recombinant gene expression in M9 medium ($\mu$ = 0.40–0.43 $h^{-1}$, S6A Fig). High-level synthesis of CYP154C5 and AlkL was detected when expected (S6B Fig). A 39 kDa protein band intensified after induction and likely represents PDR, as this band also became visible after induction with a plasmid carrying only the PDR and Pd genes (pACYC-camAB, S9 Fig). The rather small Pd (11 kDa) was not detectable in the SDS-PAGE setup applied. In accordance with previous studies summarized in Bertelmann *et al.* [17], specific progesterone hydroxylation rates were stable over 2 h but very low (~1 U $g_{CDW}^{-1}$, Fig 7). Since the presence of AlkL did not improve CYP154C5-based whole-cell steroid conversion rates, we tested, if AlkL at all can facilitate progesterone uptake by applying KSA14m to catalyze this reaction as previously reported [16]. *E. coli* cells equipped with KSA14m and AlkL indeed displayed 4.2-fold higher initial product formation rates (up to 14 U $g_{CDW}^{-1}$) than cells without AlkL (S7 Fig). Thus, an uptake limitation only becomes relevant, if high intracellular steroid conversion activities are established, which was not the case with CYP154C5.

Maintaining a large plasmid and expressing 4 genes from one single operon can cause expression issues and hamper functional expression. Consequently, genes were located on two

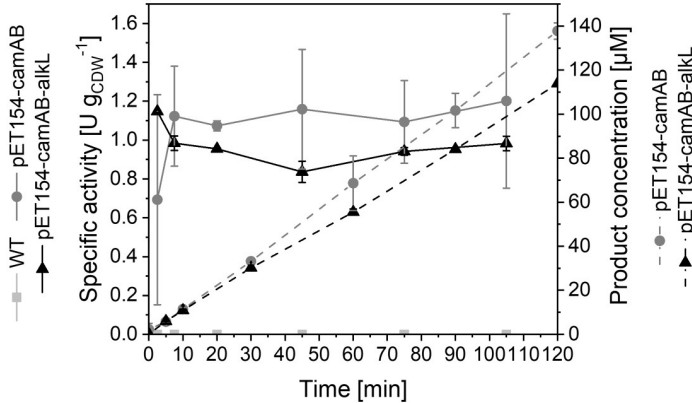

**Fig 7. Resting-cell biotransformations with *E. coli* BL21-Gold(DE3) cells carrying pETM11 with *cyp154c5*, *camA*, *camB* (and optionally *alkL*)—One plasmid strategy.** Time courses of specific progesterone hydroxylation activities (solid lines) and product concentrations (dashed lines) are depicted. Bacteria were cultivated in M9 medium containing 0.5% (w/v) glucose. Resting cells were prepared 5 h after induction with 0.1 mM IPTG and applied as described in Materials and Methods. Average values and standard deviations of biological duplicates are presented.

**Table 2. Specific growth rates and specific activities of *E. coli* BL21-Gold(DE3) strains carrying genes encoding alternative CYP450s[a].**

| Plasmid backbone | pCOM10_tac | | | | pETM11 | | | |
|---|---|---|---|---|---|---|---|---|
| Genes | *cyp154c5* | *cyp154c5, alkL* | *cyp106a2* | *cyp106a2, alkL* | *cyp154c5* | *cyp154c5, alkL* | *cyp106a2* | *cyp106a2, alkL* |
| Resulting plasmid | ptac154 | ptac154A | ptac106 | ptac106A | pET154 | pET154A | pET106 | pET106A |
| Specific growth rate [h⁻¹] | 0.339 ± 0.002 | 0.336 ± 0.001 | 0.366 ± 0.002 | 0.346 ± 0.002 | 0.414 ± 0.001 | 0.416 ± 0.003 | 0.424 ± 0.006 | 0.412 ± 0.001 |
| Resting-cell activity [U g_{CDW}⁻¹] | 1.72 ± 0.13 (5 min) | 1.45 ± 0.03 (5 min) | no product detected for 20 h | ~45 µM detected after 20 h | 3.32 ± 0.11 (5 min) | 1.58 ± 0.01 (5 min) | no product detected for 20 h | ~50 µM detected after 20 h |

[a] Plasmids based on either pCOM10_tac or pETM11 were introduced to provide *cyp154c5* or *cyp106a2* and optionally *alkL*. All strains harbored pACYC-camAB encoding PDR and Pd necessary for electron delivery.

separate plasmids in a second approach. The respective CYP450 gene and optionally *alkL* were placed on pETM11 or pCOM10_tac (S5B Fig), leading to eight different plasmids carrying *cyp154c5* or *cyp106a2* and optionally *alkL*. As second plasmid, pACYC was equipped with *camA* and *camB* giving rise to pACYC-camAB (S5A Fig), of which the pA15 origin is compatible with both pMB1 and ColE1 of pETM11 and pCOM10_tac, respectively.

The only slight decrease of specific growth rates compared to the wild type (0.43 h⁻¹, Table 2) indicated that *E. coli* BL21-Gold(DE3) can cope with the metabolic burden imposed by the presence of two plasmids and the simultaneous expression of heterologous genes (S8 Fig). For strains carrying the *cyp106a2* gene, no or only trace amounts of product were detected after 20 h of biotransformation (Table 2). CYP106A2 was not observed on SDS gels at the expected size (45 kDa), but may be represented by a thin band at ~60 kDa which intensified after induction (S9B–S9D Fig). In contrast, SDS-PAGE analyses demonstrated clear bands for CYP154C5 (S9A, S9C and S9D Fig). AlkL was observed when expected and also bands for PDR and Pd were detected, whereas significant leaky expression occurred in most cases. Initial resting-cell activities obtained with CYP154C5 were slightly higher than those obtained with one plasmid, with pET154 enabling a 3-fold increase (Table 2). However, these testosterone hydroxylation rates were still low, not subject to AlkL-mediated improvement, and not of special interest for industrial implementation.

## Conclusion

Testosterone hydroxylation can be catalyzed at exceptionally high rates by whole-cell biocatalysts carrying the BM3 variant KSA14m and the uptake facilitator AlkL, but suffers from an inherent instability. In order to optimize biocatalyst stability, this work evaluated a wide variety of approaches proven successful for processes employing other oxygenases (Table 3). Different biocatalyst formats and enzyme variants were tested and conditions for heterologous enzyme synthesis were varied in terms of cultivation medium, inducer concentration, expression system, and microbial host. None of these approaches however improved operational stability or initial specific activity. Whereas these findings provide valuable guidance for researchers tackling similar CYP450 stability issues, an activity comparison of the BM3-based whole-cell biocatalysts with natively steroid-hydroxylating CYP450s and suitable redox partners in equivalent biocatalyst and reaction setups qualified the former as clearly superior. BM3 stabilization *via* protein engineering, especially substrate affinity/positioning, is considered to

**Table 3. Summary of strategies investigated to stabilize KSA14m-catalyzed testosterone hydroxylation using whole-cell systems and their effect on initial specific activity and stability.**

| Strategy | Aim | Effect on initial specific activity | Effect on stability of activity |
|---|---|---|---|
| Biotransformation with magnesium-deprived resting cells in a complete growth medium | fostering capacity for functional KSA14m (re-)synthesis during biotransformation by supplying nitrogen and iron | no effect | no effect |
| Presence of inducer during resting-cell biotransformation | fostering KSA14m (re-)synthesis by induction of gene transcription during biotransformation | no effect | no effect |
| Biotransformation using growing cells | efficient (re-)synthesis of cells, enzymes, and energy/redox carriers during biotransformation superior handling of biocatalysis-related stress | 3-fold reduction compared to resting cells | no effect |
| Variation of cultivation media for heterologous protein synthesis (resting-cell biotransformation) | variation in cell physiology leading to higher levels of functional KSA14m | level after cultivation in different media:<br>M9 > RB > M9* | no effect |
| Variation of inducer concentration for heterologous protein synthesis (resting-cell biotransformation) | reduction of expression to achieve:<br>• reduction of metabolic burden<br>• higher share of functional enzyme (protein folding, heme incorporation)<br>• lower KSA14m levels decreasing cofactor demand and possibly uncoupling | lowest inducer concentrations enabled highest activities | no effect |
| Variation of expression system for heterologous protein synthesis (resting-cell biotransformation) | | level with different regulatory systems:<br>$lacI$-$P_{T7}$ > $lacI^q$-$P_{tac}$ > $lacI$-$P_{lacUV5}$ > $alkS$-$P_{alkB}$ | no effect, except for slight stabilization with $lacI$-$P_{lacUV5}$ regulatory system at low biomass density |
| KSA14m engineering (resting-cell biotransformation) | reduction of possible uncoupling-related instability | no effect, except for 1.5-fold reduction with M82A | no improvement, fastest decrease with L188P variant |
| *Pseudomonas taiwanensis* as host organism (resting-cell biotransformation) | supporting functional KSA14m synthesis by intrinsic heme maturation apparatus | significantly lower | not investigated |

remain as the most promising approach given the large and fast developing tool set for structure prediction and modification.

## Supporting information

**S1 Table. Primer sequences and PCR templates used for the construction of the plasmids used in this study.**
(PDF)

**S2 Table. Components and settings of site-directed mutagenesis PCR to introduce M82A and L188P amino acid exchanges into KSA14m.**
(PDF)

**S3 Table. Composition and preparation of minimal media used in this study.**
(PDF)

**S1 Fig. Genetic constructs applied for co-expression of the genes encoding the 6xHis-tagged BM3 variant KSA14m and the hydrophobic outer membrane protein AlkL in a bicistronic operon under control of different expression systems.** Heterologous gene expression was either realized via **(A)** the T7 promoter ($P_{T7}$), **(B)** the alkB promoter ($P_{alkB}$), **(C)** the lacUV5 promoter ($Plac_{UV5}$), or **(D)** the tac promoter ($P_{tac}$) based on either LacI$^q$ (A, C, D) or AlkS (B) as regulators. The creation of plasmids pETM11-ksa14m-alkL (A), palk-ksa14m-alkL (B), plac-ksa14m-alkL (C), and ptac-ksa14m-alkL (D) is explained in S1 Table.
(JPG)

**S2 Fig. Courses of product formation during biocatalytic testosterone hydroxylation with resting and growing *E. coli* BL21-Gold(DE3) cells.** Microorganisms were cultivated in M9

medium containing 0.5% (w/v) glucose and heterologous gene expression was induced with 0.1 mM IPTG. Biotransformations were conducted as described in the Materials and Methods Section. Average values and standard deviations of two biological replicates are shown.
(TIF)

**S3 Fig. SDS-PAGE analyses for *E. coli* BL21-Gold(DE3) strains producing KSA14m (119 kDa, black arrows) and AlkL (23 kDa, grey arrows) under control of different gene expression systems (see S1 Fig) and with different inducer concentrations.** Microorganisms were cultivated in M9 medium containing 0.5% (w/v) glucose and heterologous gene expression was induced in the early exponential phase. **(A)** AlkS-$P_{alkB}$-based regulatory system, induced with different DCPK concentrations. **(B)** LacI-$P_{lacUV5}$-based expression system, induced with different IPTG concentrations. **(C)** LacI$^q$-$P_{tac}$-based expression system, induced with different IPTG concentrations. **(D)** LacI-$P_{T7}$-based regulatory system, either uninduced or induced with different IPTG concentrations.
(JPG)

**S4 Fig. Growth of *E. coli* BL21-Gold(DE3) strains carrying pETM11 equipped with *alkL* and either the original *ksa14m* gene or variants thereof (*ksa14mutations*, *ksa14m_M82A*, *ksa14m_L188P*).** Bacterial cells were grown in M9 medium containing 0.5% (w/v) glucose and heterologous gene expression was induced with 0.1 mM IPTG in the early exponential phase. Average values and standard deviations of two biological replicates are given.
(TIF)

**S5 Fig. Genetic constructs applied for co-expression of genes encoding alternative steroid hydroxylation systems.** Necessary genes encoded either CYP154C5 or CYP106A2 (*cyp*), redox partner proteins (*camA* for putidaredoxin reductase, PDR, and *camB* for putidaredoxin, Pd), as well as the hydrophobic outer membrane protein AlkL (*alkL*). **I)** One plasmid strategy: one vector for the synthesis of 6xHis-tagged CYP154C5, PDR, Pd, and AlkL under control of the LacI-$P_{T7}$-based expression system. **II)** Two plasmid strategy: *cyp*/*alkL* and *camA*/*camB* were expressed from two separate plasmids. Final bacterial strains always contained plasmid **A** (pACYC-camAB) carrying *camA* and *camB* under control of two separate T7 promoters. In addition, the cells harbored either plasmid **B1** or **B2** carrying the respective *cyp* genes and *alkL* under control of $P_{T7}$ (B1) or $P_{tac}$ (B2). Plasmid construction is detailed in S1 Table.
(JPG)

**S6 Fig. Performance of *E. coli* BL21-Gold(DE3) cells carrying pETM11 equipped with *cyp154c5*, *camA*, *camB*, and optionally *alkL* (one plasmid strategy).** Microorganisms were cultivated in M9 medium containing 0.5% (w/v) glucose and heterologous gene expression was induced with 0.1 mM IPTG. **(A)** Growth of recombinant strains compared to the wild type (WT). Average values and standard deviations of biological duplicates are presented. **(B)** SDS-PAGE analysis of heterologous protein levels achieved with the respective strains after different times of induction. Expected proteins are marked with arrows (CYP154C5, 45 kDa; PDR, 45 kDa; AlkL, 23 kDa; Pd, 11 kDa).
(JPG)

**S7 Fig. Comparison of initial specific progesterone hydroxylation activities of *E. coli* BL21-Gold(DE3) cells harboring pETM11 equipped with *ksa14m* with or without *alkL*.** Resting cells were obtained and used for progesterone hydroxylation assays as described in the Materials and Methods section. As the expected products 2β- and 16β-hydroxyprogesterone (Kille *et al*., 2011) were not available for calibration curve generation, steroid concentrations were quantified based on standard curves for the similar compound 16α-hydroxyprogesterone

and used for the calculation of initial product formation rates (5 min) of strains with or without the hydrophobic outer membrane pore AlkL (biological duplicates).
(TIF)

**S8 Fig. Growth of *E. coli* BL21-Gold(DE3) cells carrying plasmids equipped with *cyp154c5*, *camA*, *camB*, and *alkL* (two plasmid strategy).** All strains contained pACYC-camAB and a second plasmid carrying *cyp154c5* or *cyp106a2* together with or without *alkL* on pCOM10_tac **(A)** or pETM11 **(B)**. Recombinant bacteria were cultivated in M9 medium supplemented with 0.5% (w/v) glucose and heterologous gene expression was induced with 0.1 mM IPTG. Data points represent average values and standard deviations of two biological replicates.
(JPG)

**S9 Fig. SDS-PAGE analyses of induced *E. coli* BL21-Gold(DE3) cells carrying genes encoding CYP154C5 or CYP106A2 (45 kDa), PDR (45 kDa), Pd (11 kDa), and optionally AlkL (23 kDa) separately on two vectors (two plasmid strategy).** Bacteria were grown in M9 medium containing 0.5% (w/v) glucose and heterologous gene expression was induced with 0.1 mM IPTG. Expected proteins are marked with arrows in the respective gel pictures.
(JPG)

**S1 Raw images. Collection of all original SDS gel images contained in the manuscript's main and supplementary figures.**
(PDF)

**S1 Raw data. Minimal data set.**
(XLSX)

## Acknowledgments

The authors acknowledge the use of the facilities of the Centre for Biocatalysis (MiKat) at the Helmholtz Centre for Environmental Research, which is supported by the European Regional Development Funds (EFRE, Europe funds Saxony) and the Helmholtz Association.

## Author Contributions

**Conceptualization:** Carolin Bertelmann, Bruno Bühler.

**Data curation:** Carolin Bertelmann.

**Formal analysis:** Carolin Bertelmann.

**Funding acquisition:** Bruno Bühler.

**Investigation:** Carolin Bertelmann, Bruno Bühler.

**Methodology:** Carolin Bertelmann, Bruno Bühler.

**Project administration:** Carolin Bertelmann, Bruno Bühler.

**Resources:** Bruno Bühler.

**Supervision:** Bruno Bühler.

**Validation:** Carolin Bertelmann, Bruno Bühler.

**Visualization:** Carolin Bertelmann, Bruno Bühler.

**Writing – original draft:** Carolin Bertelmann.

**Writing – review & editing:** Carolin Bertelmann, Bruno Bühler.

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
