## [Decision Letter · Decision Letter 0]

16 Jul 2024

PONE-D-24-24922Strategies found not to be suitable for stabilizing high steroid hydroxylation activities of CYP450 BM3-based whole-cell biocatalystsPLOS ONE

Dear Dr. Bühler,

Thank you for submitting your manuscript to PLOS ONE. After careful consideration, we feel that it has merit but does not fully meet PLOS ONE’s publication criteria as it currently stands. Therefore, we invite you to submit a revised version of the manuscript that addresses the points raised during the review process.

We look forward to receiving your revised manuscript.

Kind regards,

Dirk Tischler

Academic Editor

PLOS ONE

Journal Requirements:

   "The study was financially supported by Bayer AG (no grant number).

Website: https://www.bayer.com

B.B. aquired the funding and CB was paid from the funding.

Bayer AG released the submitted paper for publication."

Additional Editor Comments:

Dear Bruno

you will see the comments from the reviewers are supportive and I am looking forward to read the revision. And, as it was hard to appoint reviewers in time and thus we reached holiday saison I understand that it may take time. Still I hope you will improve and send a next version to make a decision.

Sincerely

Dirk

Reviewers' comments:

Reviewer's Responses to Questions

**Comments to the Author**

1. Is the manuscript technically sound, and do the data support the conclusions?

Reviewer #1: Partly

Reviewer #2: Yes

Reviewer #3: Yes

2. Has the statistical analysis been performed appropriately and rigorously? 

Reviewer #1: N/A

Reviewer #2: N/A

Reviewer #3: Yes

3. Have the authors made all data underlying the findings in their manuscript fully available?

Reviewer #1: No

Reviewer #2: Yes

Reviewer #3: Yes

4. Is the manuscript presented in an intelligible fashion and written in standard English?

Reviewer #1: Yes

Reviewer #2: Yes

Reviewer #3: Yes

5. Review Comments to the Author

Reviewer #1: In their manuscript “Strategies found not to be suitable for stabilizing high steroid hydroxylation activities of CYP450 BM3-based whole-cell biocatalysts”, Bertelmann and Bühler explore various strategies aimed to increase the stability of whole-cell steroid-hydroxylation by Escherichia coli equipped with heterologous cytochrome P450 monooxygenases (CYP450s). Approaches include biotransformation media/buffers with different nutrient compositions, growing cells vs. resting cells, different growth media, different expression systems (prompters, inducers, inducer concentrations), protein engineering, using Pseudomonas as a host, and testing different CYP450 enzymes.

In general, although the authors state that they did not achieve a better system (but see my comment on that below), the work seems to be carefully conducted and is of interest for applied studies.

However, the manuscript should be presented in a more reader-friendly way. The authors assume a lot of knowledge in the abstract and the introduction (see comments).

Also, although the results might indeed be a guideline for further studies, the article is scientifically somewhat unsatisfying, because likely reasons for decreasing activity are discussed, but no data are shown. For example, what about levels of O2 in the biotransforming cell cultures, heme content in CYP450s, or NAD(P)H:NAD(P)+ ratios? If there are values known from literature, I suggest mentioning them, perhaps in the introduction (see my comment Nr. 2).

Finally, I do not understand why the authors did not proceed with, and did not do more experiments on, the plac system (Fig. 4B), and particularly the 0.1 mM IPTG / 0.1 g CDW setup (see my comment Nr. 6). This should be commented on.

In summary, in my opinion, the article could be published without further experiments, but with a thorough editing of the text.

Comments in detail:

1. The abstract is a bit confusing for readers not familiar with this field and the preceding study, respectively. Perhaps the authors may shorten the description of results to make some room for briefly summarizing the former setup that they aimed to optimize.

2. The same holds true for the introduction. It would benefit from introducing CYP450s a bit, such as specifying their cofactors and redox partners, or what is meant by “multi-component nature” (line 48). Also, the previous system should be introduced better, e.g., what is meant by “self-sufficient” (line 51), what is uncoupling (line 62; although this is explained later), what is the KSA14m variant (line 64)… It would also help the readers a lot if you introduced possible optimization steps (or bottlenecks) before delving into the results.

3. Lines 167-168: When the cells are resuspended in potassium phosphate buffer, they are depleted for every nutrient (except K, P; and glucose). Could this influence catalysis and/or stability? For example, what happens with a CYP450 when iron becomes limiting? Why did you not use M9 without nitrogen?

4. Results: Specific activity does not correlate with the enzyme level (e.g., Fig. 1A; Fig. 2). Have you ever purified the enzyme from the cells and checked its cofactor load? Or else, could oxygen be a limiting factor? Since you add glucose, I suppose that the cells have a high respiration rate? The results obtained with low cell concentrations you present in Fig. 4 (and accompanying text) suggest that O2 might not be the limiting factor, but did you ever measure O2 levels in your biotransformation cultures? If you analyzed these things previously, it would be good to mention that. If not, I do not request these analyses for publication of this manuscript, but they might be worth a try in the future.

5. Depending on your response to 4., it might also be worth mentioning that E. coli strains with optimized heme uptake or production have been generated.

6. Also results, Fig. 4: why did you not follow up on the plac system (Fig. 4B), and the 0.1 mM IPTG / 0.1 g CDW setup in particular? It seems to me that it stabilized, although on a somewhat lower than maximally achievable level. Still, did you ever measure activity at later time points? Did you quantify product yields?

7. Do the results shown in Fig. 5B not argue for enzyme destabilization independent from substrate / product concentrations? This could be mentioned much earlier in the manuscript to avoid questions piling up.

8. Lines 495-496: It would be nice for the reader if you would explain what fhuA and todX mean. Also, I assume you want to refer to Fig. 6, not 4.

9. Fig. 6 and results with Pseudomonas, respectively: Obviously, CYP350 levels are not very high in this strain. Did you consider changing expression vectors before dismissing this species completely?

10. Line 545: To evaluate this system, it is important to know the putidaredoxin levels. At least for selected conditions, this should be done, e.g. by higher-percentage gels and/or a Tris/Tricine gel.

11. Conclusions: The manuscript is somewhat overwhelming. I suggest showing a simplified summarizing table.

Minor:

12. Fig. 2A, left panel: the product symbols look like the testosterone symbols after 1 h.

13. Results, Fig. 4 and Fig. 5: Please indicate which resting cell system was applied.

14. Line 440 and line 446: Is it position 87 or 82? Please specify.

15. Table 2 is hard to understand; I’d draw lines between the rows in the lower part.

Reviewer #2: Previously, the same group investigated a whole-cell E. coli-based biocatalyst co-expressing the outer membrane pore AlkL and the P450 BM3 variant KSA14m capable of hydroxylating testosterone primarily to the 15-b-hydroxylated product. During biotransformation, resting cells lose activity due to instability of BM3 monooxygenase. In the present study, the authors tested a number of approaches and varied several factors in an attempt to increase enzyme stability during the process, which could increase the duration of biotransformation and thus lead to higher overall product concentrations. Among other things, they compared resting and growing E. coli cells, varied nutritional conditions during gene expression, different media, promoters, and inducer concentrations. None of these changes resulted in improved cell activity or increased stability of BM3 KSA14m during biotransformation. Interestingly, culture medium and expression strategies influence the initial whole cell activity but not its stability. Furthermore, two additional stabilizing mutations were introduced into the monooxygenase gene, which ultimately did not change the product titer. The use of Pseudomonas instead of E. coli did not improve productivity. Two other steroid hydroxylating P450s, CYP154C5 and CYP106A2, were cloned in E. coli together with the two redox partner genes. While CYP106A2 could not be expressed at all, the biotransformation of testosterone by cells expressing CYP154C5 was found to be no more productive than cells with BM3 KSA14m.

In general, although no improvement could be achieved within this study, the "negative" results can be very helpful for further studies on optimization of whole cell biocatalysts for oxygenation reaction.

I recommend to publish the paper after some revisions:

1. As shown in Figure 2, growing cells continued to express KSA14m even after 24 h of biotransformation (compared to resting cells), but this did not lead to higher biocatalyst productivity. This was explained by a stronger competition of enzyme synthesis and metabolic demands of the reaction with the demands of cellular maintenance and biomass formation. It seems that cell activity (independently if growing or resting) is dependent on the P450 concentration per 1 g cww, as clearly shown in Fig. 2a and Fig. 2b. In the case when the induction was done simultaneously with substrate addion, the specific cell activity first increases with increasing P450 concentration (see the corresponding SDS gel) and then decreases with complete loss after 3.5 h. It would be interesting to measure P450 concentration (using CO difference spectrum). It will not change the situation with the loss of activity after a certain time of biotransformation, but it would allow a more concrete explanation that specific cell activities in growing cells were lower (even at the very beginning of the biotransformation) compared to the resting cells due to lower P450 concentration per cww.

2. It would be helpful for the readers if the genetic construct- AlkL and KSA14m expressed from the same operon – will be described in the introduction section.

3. L. 316-317: “Obviously, KSA14m experienced a similar inactivation in growing as in resting cells,which could not be alleviated by enzyme resynthesis in growing cells”. Compared to resting cells in which the P450 band disappeared after 24 biotransformation, the band corresponding to the P450 remains in growing cells. Could it be that the loss of activity is not only due to protein instability, but due to the loss of heme or FAD or FMN (the prosthetic groups of the reductase domain of P450 BM3)? None of these aspects have been checked in this study, but it is importnat to at least mention them in the discussion.

4. L. 375-377: Please mention the promoters used in this study and their strength compared to each other to make clearer their choice for this study.

5. Was CYP106A2 not active in whole cells or was cloning unsuccessful?

6. Fig.2 In both cases, 700 mkM product concertation was achieved after 2.5 – 4 h starting with 1 mM (=1000 mkM) testosterone. Was the productivity of resting cells used at the same cell density higher?

Reviewer #3: The manuscript presents negative results, however of important biotechnology field. The authors performed many rigorous, well documented and analyzed experiments but were not able to reach desired stability of whole cell oxidation of steroids.

I have only a few recomendations:

1. I am missing a discussion which would discuss more widely the impacts of the findings both to the scientific field as well as to the biotechnology praxis.

2. English is of high level, except of a few wrong articles:

- "the" missing in front of the "most" on several places.

- "a first approach" should be "the" (line 535).

6. PLOS authors have the option to publish the peer review history of their article (what does this mean?). If published, this will include your full peer review and any attached files.

Reviewer #1: No

Reviewer #2: No

Reviewer #3: **Yes: **Josef Trögl

---

## [Author Response · Author response to Decision Letter 0]

19 Aug 2024

Response to editor

Journal Requirements:

We have revised the manuscript and supporting data to meet the PLOS ONE style requirements according to the templates and the information given by the website.

We have removed funding information from the Acknowledgements section of the manuscript (line 653).

 "The study was financially supported by Bayer AG (no grant number).

Website: https://www.bayer.com

B.B. aquired the funding and CB was paid from the funding.

Bayer AG released the submitted paper for publication."

The funders participated in project design and approved the submission of the manuscript, but had no role in study design, data collection and analysis, or preparation of the manuscript.

The minimal data set now is provided in the Supporting Information file “S2_raw_data.xlsx”.

The first manuscript included two appearances of this phrase:

In line 539, the strain Pseudomonas taiwanensis VLB120_Strep containing the plasmid ptac-ksa14m-alkL did not grow in the LB preculture unlike the other Pseudomonas strains tested. This was a visual observation not based on measured data, but the respective comment was noted in the laboratory journal. Therefore, we have removed the phrase from that sentence.

In line 547, we have added the reference to the corresponding raw image SDS gel.

We confirm that the reference list is complete and correct. During revision, following the reviewer recommendations, 16 references have been added to the list (listed in order of appearance):

Bernhardt R. Cytochromes P450 as versatile biocatalysts. J Biotechnol. 2006;124(1):128-145. doi: 10.1016/j.jbiotec.2006.01.026

Ortiz de Montellano PR. Cytochrome P450: structure, mechanism, and biochemistry. 3 ed. New York: Springer Verlag; 2005

Urlacher VB, Eiben S. Cytochrome P450 monooxygenases: perspectives for synthetic application. Trends Biotechnol. 2006;24(7):324-330. doi: 10.1016/j.tibtech.2006.05.002

Murdock D, Ensley BD, Serdar C, Thalen M. Construction of metabolic operons catalyzing the de novo biosynthesis of indigo in Escherichia coli. Biotechnology (N Y). 1993;11(3):381-386. doi: 10.1038/nbt0393-381

Guengerich FP. Rate-limiting steps in cytochrome P450 catalysis. Biol Chem. 2002;383(10):1553-1564. doi: 10.1515/BC.2002.175

Narhi LO, Fulco AJ. Characterization of a catalytically self-sufficient 119,000-dalton cytochrome P-450 monooxygenase induced by barbiturates in Bacillus megaterium. J Biol Chem. 1986;261(16):7160-7169. doi: 10.1016/S0021-9258(17)38369-2

Narhi LO, Fulco AJ. Identification and characterization of two functional domains in cytochrome P-450BM-3, a catalytically self-sufficient monooxygenase induced by barbiturates in Bacillus megaterium. J Biol Chem. 1987;262(14):6683-6690. doi: 10.1016/S0021-9258(18)48296-8

Tegel H, Ottosson J, Hober S. Enhancing the protein production levels in Escherichia coli with a strong promoter. FEBS J. 2011;278(5):729-739. doi: 10.1111/j.1742-4658.2010.07991.x

Terpe K. Overview of bacterial expression systems for heterologous protein production: from molecular and biochemical fundamentals to commercial systems. Appl Microbiol Biotechnol. 2006;72(2):211-222. doi: 10.1007/s00253-006-0465-8

Makart S, Heinemann M, Panke S. Characterization of the AlkS/PalkB-expression system as an efficient tool for the production of recombinant proteins in Escherichia coli fed-batch fermentations. Biotechnol Bioeng. 2007;96(2):326-336. doi: 10.1002/bit.21117

Panke S, Wubbolts MG, Schmid A, Witholt B. Production of enantiopure styrene oxide by recombinant Escherichia coli synthesizing a two-component styrene monooxygenase. Biotechnol Bioeng. 2000;69(1):91-100. doi: 10.1002/(sici)1097-0290(20000705)69:1<91::aid-bit11>3.0.co;2-x

Wang Y, Rawlings M, Gibson DT, Labbé D, Bergeron H, Brousseau R, et al. Identification of a membrane protein and a truncated LysR-type regulator associated with the toluene degradation pathway in Pseudomonas putida F1. Mol Gen Genet. 1995;246(5):570-579. doi: 10.1007/BF00298963

Ruff AJ, Arlt M, van Ohlen M, Kardashliev T, Konarzycka-Bessler M, Bocola M, et al. An engineered outer membrane pore enables an efficient oxygenation of aromatics and terpenes. J Mol Catal B Enzym. 2016;134:285-294. doi: 10.1016/j.molcatb.2016.11.007

Lee MJ, Kim H-J, Lee J-Y, Kwon AS, Jun SY, Kang SH, et al. Effect of gene amplifications in porphyrin pathway on heme biosynthesis in a recombinant Escherichia coli. J Microbiol Biotechnol. 2013;23(5):668-673. doi: 10.4014/jmb.1302.02022

Ge J, Wang X, Bai Y, Wang Y, Wang Y, Tu T, et al. Engineering Escherichia coli for efficient assembly of heme proteins. Microb Cell Fact. 2023;22:59. doi: 10.1186/s12934-023-02067-5

Hu B, Yu H, Zhou J, Li J, Chen J, Du G, et al. Whole-cell P450 biocatalysis using engineered Escherichia coli with fine-tuned heme biosynthesis. Adv Sci. 2023;10(6):e2205580. doi: 10.1002/advs.202205580

 

Response to reviewers

Reviewer #1: In their manuscript “Strategies found not to be suitable for stabilizing high steroid hydroxylation activities of CYP450 BM3-based whole-cell biocatalysts”, Bertelmann and Bühler explore various strategies aimed to increase the stability of whole-cell steroid-hydroxylation by Escherichia coli equipped with heterologous cytochrome P450 monooxygenases (CYP450s). Approaches include biotransformation media/buffers with different nutrient compositions, growing cells vs. resting cells, different growth media, different expression systems (prompters, inducers, inducer concentrations), protein engineering, using Pseudomonas as a host, and testing different CYP450 enzymes.

In general, although the authors state that they did not achieve a better system (but see my comment on that below), the work seems to be carefully conducted and is of interest for applied studies.

However, the manuscript should be presented in a more reader-friendly way. The authors assume a lot of knowledge in the abstract and the introduction (see comments).

Also, although the results might indeed be a guideline for further studies, the article is scientifically somewhat unsatisfying, because likely reasons for decreasing activity are discussed, but no data are shown. For example, what about levels of O2 in the biotransforming cell cultures, heme content in CYP450s, or NAD(P)H:NAD(P)+ ratios? If there are values known from literature, I suggest mentioning them, perhaps in the introduction (see my comment Nr. 2).

Finally, I do not understand why the authors did not proceed with, and did not do more experiments on, the plac system (Fig. 4B), and particularly the 0.1 mM IPTG / 0.1 g CDW setup (see my comment Nr. 6). This should be commented on.

In summary, in my opinion, the article could be published without further experiments, but with a thorough editing of the text.

Comments in detail:

1. The abstract is a bit confusing for readers not familiar with this field and the preceding study, respectively. Perhaps the authors may shorten the description of results to make some room for briefly summarizing the former setup that they aimed to optimize.

We have revised the abstract according to the reviewer’s suggestion (lines 18-33, 40).

2. The same holds true for the introduction. It would benefit from introducing CYP450s a bit, such as specifying their cofactors and redox partners, or what is meant by “multi-component nature” (line 48). Also, the previous system should be introduced better, e.g., what is meant by “self-sufficient” (line 51), what is uncoupling (line 62; although this is explained later), what is the KSA14m variant (line 64)… It would also help the readers a lot if you introduced possible optimization steps (or bottlenecks) before delving into the results.

We agree with this comment and thus have revised the introduction as recommended (lines 44-45, 47-60, 64-71, 74-81, 89, 92). The purpose of this study is now mentioned more clearly in the introduction (lines 84-86) as well as in the beginning of the Results&Discussion section (lines 240-242). Possible optimization steps/bottlenecks are introduced in lines 98-108. 

3. Lines 167-168: When the cells are resuspended in potassium phosphate buffer, they are depleted for every nutrient (except K, P; and glucose). Could this influence catalysis and/or stability? For example, what happens with a CYP450 when iron becomes limiting? Why did you not use M9 without nitrogen?

The lack or depletion of nutrients can indeed affect catalysis as well as stability due to their respective demand in the whole-cell biocatalyst’s metabolism. In this study, the standard biotransformation setup with resting (i.e., non-growing but metabolically active) cells included potassium, phosphate, and glucose as sole components, as it also has broadly been reported in literature. The lack of a nitrogen source thereby is used to achieve a growth limitation, which transfers the cells into a resting, but metabolically active state. In this state, competition of metabolic demands of the biocatalytic reaction with cellular maintenance and biomass formation is reduced. This often allows superior specific biotransformation rates of resting cells compared to growing cells (as also observed in this study, Figs. 1 and 2), ultimately resulting in increased product yields on energy source as well as facilitated reusability and downstream processing. However, omitting nitrogen from the biotransformation medium implies a down-regulation of biosynthetic capacities, especially the efficient (re )synthesis of enzymes and energy/redox carriers, which can affect the operational stability during the biotransformation (explained in lines 251-263, 277-283). This is the reason why we tested M9 without magnesium to exclude effects of other nutrients including nitrogen or iron. Similar activities were observed. Just deprivation of magnesium still allows some growth as shown in an earlier study, in which M9 medium without magnesium allowed better performance than M9 medium without nitrogen, whereas the latter did not result in differences compared to Kpi buffer with glucose (Willrodt C, Hoschek A, Bühler B, Schmid A, Julsing MK. Decoupling production from growth by magnesium sulfate limitation boosts de novo limonene production. Biotechnol Bioeng. 2016;113(6):1305-1314. doi: 10.1002/bit.25883). This information was added (lines 263-267).

4. Results: Specific activity does not correlate with the enzyme level (e.g., Fig. 1A; Fig. 2). Have you ever purified the enzyme from the cells and checked its cofactor load? Or else, could oxygen be a limiting factor? Since you add glucose, I suppose that the cells have a high respiration rate? The results obtained with low cell concentrations you present in Fig. 4 (and accompanying text) suggest that O2 might not be the limiting factor, but did you ever measure O2 levels in your biotransformation cultures? If you analyzed these things previously, it would be good to mention that. If not, I do not request these analyses for publication of this manuscript, but they might be worth a try in the future.

We initially aimed to purify the enzyme based on the 6x-His-tag at the N-terminal end of the enzyme for a thorough evaluation of its specific kinetic properties and uncoupling behavior (i.e., uncoupling of NADPH oxidation and substrate hydroxylation). Our attempts however were not successful due to impurity of the final eluates, which may severely influence the respective experimental outcome.

Regarding the measurement of oxygen in the biotransformation cultures: Oxygen levels were not measured for the experiments presented in this manuscript and therefore no data is available in this regard. We agree that oxygen-dependent reactions in whole-cell biocatalysis can be limited by oxygen mass transfer, especially in closed systems. In our study, assays featured a volume ratio of liquid : headspace of 1 : 11, and, for reaction time course analysis, tubes were opened regularly for sampling. Moreover, in our previous study, we investigated oxygen as limiting factor by varying biomass co

---

## [Editor Report · Decision Letter 1]

22 Aug 2024

Strategies found not to be suitable for stabilizing high steroid hydroxylation activities of CYP450 BM3-based whole-cell biocatalysts

PONE-D-24-24922R1

Dear Dr. Bühler,

We’re pleased to inform you that your manuscript has been judged scientifically suitable for publication and will be formally accepted for publication once it meets all outstanding technical requirements.

Kind regards,

Dirk Tischler

Academic Editor

PLOS ONE

Additional Editor Comments (optional):

Dear Prof. Bühler

the changes made improved the manuscript as suggested by the three reviewers and hence I accept the revised version.

Sincerely!
---

## [Editor Report · Acceptance letter]

27 Aug 2024

PONE-D-24-24922R1 

PLOS ONE

Dear Dr. Bühler, 

I'm pleased to inform you that your manuscript has been deemed suitable for publication in PLOS ONE. Congratulations! Your manuscript is now being handed over to our production team.

Kind regards, 

on behalf of

Asst. Prof. Dr. Dirk Tischler 

Academic Editor

PLOS ONE